# The embryonic role of juvenile hormone in the firebrat, *Thermobia domestica,* reveals its function before its involvement in metamorphosis

James W Truman[1,2]*, Lynn M Riddiford[1,2], Barbora Konopova[3,4], Marcela Nouzova[5], Fernando G Noriega[6,7], Michelle Herko[1]

[1]Friday Harbor Laboratories, University of Washington, Friday Harbor, United States; [2]Department of Biology, University of Washington, Seattle, United States; [3]Department of Zoology, Faculty of Science, University of South Bohemia, Ceske Budejovice, Czech Republic; [4]Institute of Entomology, Biology Centre of the Czech Academy of Sciences, Ceske Budejovice, Czech Republic; [5]Institute of Parasitology, Biology Centre of the Czech Academy of Sciences, Ceske Budejovice, Czech Republic; [6]Department of Biological Sciences and BSI, Florida International University, Miami, United States; [7]Department of Parasitology, Faculty of Science, University of South Bohemia, Ceské Budejovice, Czech Republic

*For correspondence:
jwt@uw.edu

Competing interest: The authors declare that no competing interests exist.

## Abstract

To gain insights into how juvenile hormone (JH) came to regulate insect metamorphosis, we studied its function in the ametabolous firebrat, *Thermobia domestica*. Highest levels of JH occur during late embryogenesis, with only low levels thereafter. Loss-of-function and gain-of-function experiments show that JH acts on embryonic tissues to suppress morphogenesis and cell determination and to promote their terminal differentiation. Similar embryonic actions of JH on hemimetabolous insects with short germ band embryos indicate that JH's embryonic role preceded its derived function as the postembryonic regulator of metamorphosis. The postembryonic expansion of JH function likely followed the evolution of flight. Archaic flying insects were considered to lack metamorphosis because tiny, movable wings were evident on the thoraces of young juveniles and their positive allometric growth eventually allowed them to support flight in late juveniles. Like in *Thermobia*, we assume that these juveniles lacked JH. However, a postembryonic reappearance of JH during wing morphogenesis in the young juvenile likely redirected wing development to make a wing pad rather than a wing. Maintenance of JH then allowed wing pad growth and its disappearance in the mature juvenile then allowed wing differentiation. Subsequent modification of JH action for hemi- and holometabolous lifestyles are discussed.

## eLife assessment

This **important** study presents findings regarding the role of Juvenile Hormone in development and cell differentiation in the ametabolous insect Thermobia domestica, providing an in-depth analysis of JH's roles in a member of this basally branching group. The evidence supporting the claims of the authors is **convincing**, drawing on a broad range of approaches and variety of experimental techniques. While the interpretation of this work in the wider context - its relevance for the evolution of metamorphosis - is in some places somewhat speculative, the work will be of interest to evolutionary developmental biologists studying the evolution of metamorphosis, and the evolution of insects in general.

## Introduction

Insects first appeared about 400 mya as a group of direct developing arthropods whose hatchlings were miniature versions of the adult except that they lacked the sexual specializations needed for reproduction. Starting from this ametabolous life history, insects underwent series of developmental innovations that resulted in their becoming the dominant meso-sized group of animals in most terrestrial and freshwater environments. The first life history change followed the evolution of wings and powered flight (*Kukalova-Peck, 1978*; *Belles, 2019*). In this hemimetabolous pattern, the hatchling lacked wings as well as genitalia, and these developed as immobile pads during juvenile growth and transformed into functional structures during the molt to the adult. The subsequent step from this 'incomplete metamorphosis' to complete metamorphosis involved a redirection of embryogenesis to produce a modified larval form that was adapted to feeding and growth. The transition to the adult form then occurred through a transitional stage, the pupa, thereby establishing the three-part life history diagnostic of the 'complete metamorphosis' exhibited by holometabolous insects (reviews: *Belles, 2020b*; *Jindra, 2019*; *Truman and Riddiford, 2002*; *Truman and Riddiford, 2019*). Classically, juvenile hemimetabolous stages were called nymphs and juvenile holometabolous stages were called larvae. Although this terminology is somewhat controversial, we will use it throughout this paper.

The sesquiterpene hormone, juvenile hormone (JH), is the 'gate-keeper' for metamorphosis. Both nymphs and larvae depend on it to maintain their respective forms (*Jindra et al., 2013*; *Jindra et al., 2015*). JH has two essential functions: it is required at the start of an ecdysteroid-induced molt to maintain the insect in its current stage (*Riddiford, 1976*), and its presence during intermolt periods ensures that imaginal discs and primordia remain dormant and do not begin premature morphogenesis to the adult stage (*Truman et al., 2006*). Besides its function of suppressing metamorphosis, JH is also involved in shaping polymorphisms such as those seen in the castes of social insects (*Nijhout and Wheeler, 1982*).

The role of JH in the two ametabolous orders, the Archaeognatha (jumping bristletails) and the Zygentoma (silverfish), is poorly understood. In the latter order, JH is involved in egg production in the adult firebrat *Thermobia domestica* (*Bitsch et al., 1985*), but the only JH-related action seen in the juvenile is the suppression of scale appearance by JH application during the molt from the third to the fourth juvenile stage (*Watson, 1967*). In crustaceans, the immediate precursor to JH, methyl farnesoate (MF), regulates ovarian maturation (*Laufer et al., 1998*). Since insects and other Hexapoda (i.e. Collembola, Protura and Diplura) evolved from crustaceans (*Regier et al., 2005*; *von Reumont et al., 2012*), it has been thought that MF and JH may have originally functioned in regulating reproduction and only later became involved in development as insect life histories became more complex (*Goodman and Granger, 2005*).

Although JH treatment only mildly effects postembryonic development in *Thermobia* (*Watson, 1967*), it can severely derange embryonic development (*Rohdendorf and Sehnal, 1973*). These severe embryonic effects in an ametabolous insect suggest that the JH may initially had been an embryonic hormone and later moved into the postembryonic realm to support more complex life histories.

Our study of the JH titer in *Thermobia* shows that the highest concentrations of the hormone occur during late embryogenesis. By a detailed examination of the effects of chemicals that mimic the natural JH or block its production, respectively, we find that JH is necessary for the terminal differentiation and maturation of embryonic tissues. Associated with promoting differentiation, JH also is a potent suppressor of morphogenesis. A comparison of these embryonic functions in *Thermobia* with corresponding functions in embryos from basal hemimetabolous orders indicates that the ability of JH to switch embryonic tissues from morphogenesis to terminal differentiation was in place before the appearance of flight and of metamorphosis. The development of wings and of complex insect life histories involved the reappearance of morphogenesis during postembryonic growth. In hemimetabolous insects postembryonic morphogenesis was principally related to wing development, but in the Holometabola it involved the deferred building of the adult body plan. We think that JH reappeared in the postembryonic domain of these juveniles as a counterforce to morphogenesis so that the latter was suppressed during most of juvenile growth. The disappearance of JH then allowed morphogenesis and the subsequent formation of the adult after growth was completed.

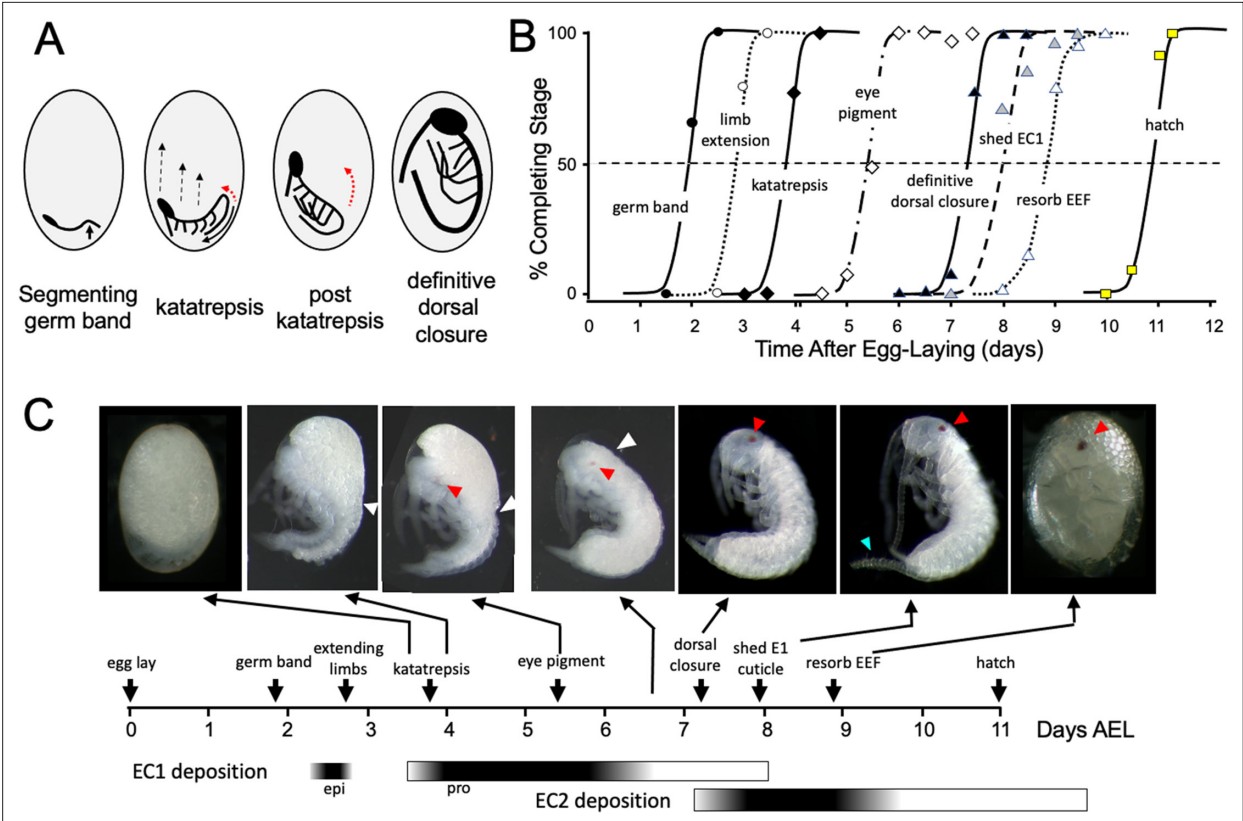

**Figure 1.** Timeline of *Thermobia domestica* embryogenesis at 37 °C. (**A**) Diagrammatic representations of important events in embryogenesis. In the segmenting germ band, the arrow indicates the invagination of the mid-abdominal region into the yolk. By the onset of katatrepsis (***Panfilio, 2008***), dorsal closure along the abdomen (red arrow) has moved the abdomen ventrally (solid black arrow) and the contraction of the serosa pulls the sides of the embryo towards the anterior pole (dashed black arrows). After katatrepsis the expanding lateral edges of the embryo displace cells of the amnion, gradually zipping up the dorsal thoracic midline (red arrow) until definitive dorsal closure is accomplished. (**B**) Summary of the progression of embryonic development at various times after egg deposition. The interception of each stage with the 50% line (dashed) is the basis for the embryonic timeline in (**C**). EEF: extraembryonic fluid. (**C**) Timeline of embryonic development based on (**B**). Micrographs show the appearance of embryos at the indicated times; first and last embryos are covered by the egg chorion. White triangle: progression of dorsal closure; red triangle: eye pigmentation; blue triangle: expanded cerci after shedding of the E1 cuticle. Times of cuticle deposition based on ***Konopová and Zrzavý, 2005***.

The online version of this article includes the following source data for figure 1:

**Source data 1.** The percent of embryos attaining a particular developmental 'milestone' as a function of the time after egg laying at 37 °C.

## Results

### The time-course of embryogenesis in *Thermobia*

The embryogenesis of *Thermobia domestica* is summarized in ***Figure 1***. At 37 °C, the time from egg deposition to hatching is about 11.5 days. A developmental timetable was constructed using 12 hr egg collections and determining the distribution of developmental stages at half day intervals from the midpoint of each collection window (see Methods). Ages are given in days after egg laying (# d AEL). As with insects from the other ametabolous order (the Archaeognatha), *Thermobia* embryos undergo short germ band development (***Jura, 1972***). An embryonic rudiment forms at the posterior pole of the egg and consists of head and thoracic segments with a terminal growth zone. As abdominal segments are progressively specified, the ventral midline invaginates into the yolk causing the abdomen to fold onto itself, thereby opposing the ventral surfaces of the anterior and posterior segments (***Figure 1A***). By about 1.5 d AEL, the embryo is evident as a cleared area at the posterior pole of the egg, and by a half day later abdomen formation is finished, and appendage buds are appearing. The dorsal surface of the embryo is open to the yolk and the embryo has an overall S-shape (***Figure 1A***).

The embryo then begins 'zipping up' its dorsal midline starting at the end of the abdomen. The dorsal closure of the abdomen pushes the abdominal tip ventrally and forward as the embryo and

its yolk assume a 'comma' shape. Growing appendage buds have reached about a third of their final length by this time. Extraembryonic membranes growing up from the lateral sides of the embryo envelop the yolk mass and then, during katatrepsis, they contract to pull the head of the embryo towards the anterior pole, and the embryo rotates to achieve a C-shape. The enclosure of the yolk by the extraembryonic membranes defines provisional dorsal closure. The embryo then begins rapid morphogenetic growth and cell determination. During this phase of organogenesis, the lateral margins of the embryonic thorax extend dorsally to completely enclose the yolk at definitive dorsal closure (about 7.5 d AEL). Having reached its full size, the embryo then undergoes terminal differentiation and maturation over the next 4 days.

Embryogenesis is accompanied by two bouts of cuticle production. The first embryonic (E1) cuticle is deposited after the formation of the segmented germ band. It contains little chitin (see below) and has a fibrous structure (*Klag, 1978*; *Konopová and Zrzavý, 2005*) which presumably allows it to stretch to accommodate subsequent embryonic growth. The second embryonic (E2) cuticle starts to be deposited around dorsal closure as the embryo reaches its full size. The E2 procuticle has the lamellar structure (*Konopová and Zrzavý, 2005*) typical of most insect cuticles (*Neville, 1975*) and serves as the cuticle of the first juvenile (J1) stage. This cuticle is shed a day after hatching with the start of the J2 stage. Feeding begins in the J3 stage.

## JH and ecdysteroid titers during embryogenesis of *Thermobia domestica*

As in most insects, *Thermobia* produces JH III (*Baker et al., 1984*). Using the liquid chromatography-mass spectrometric method of *Ramirez et al., 2020*, we measured JH III levels during *Thermobia* embryogenesis at daily intervals starting at 5 d AEL and extending through hatching and the first 10 days of postembryonic life (*Figure 2A*). Detectable levels of JH III were found at 6 d AEL, and the titer slowly ramped up until an abrupt peak on days 10 and 11. After hatching, JH III fell to low, but detectable, levels through the start of the J4 stage.

Ecdysteroid titers through embryogenesis and the early juvenile instars were measured using the enzyme immunoassay method (*Porcheron et al., 1989*). As seen in *Figure 2B*, there was a shallow, transient rise in ecdysteroid from 2.5 to 4 d AEL, associated with the time of deposition of the E1 cuticle (*Konopová and Zrzavý, 2005*). A prominent peak of 20 hydroxyecdysone (20E) then occurred around the time of dorsal closure, coordinated with the production of the E2 cuticle, and another 20E peak occurred just before hatching. We assume that this late embryonic peak is responsible for producing the J2 cuticle. The subsequent ecdysis to the J3 stage is preceded by an increase in 20E 1–2 days earlier.

*Figure 2C* summarizes the expression of some endocrine-related transcripts. Using real-time PCR (and the comparative Ct method; *Schmittgen and Livak, 2008*), we measured transcript levels at half-day intervals through the first half of embryogenesis and then at daily intervals thereafter. Transcripts for the JH receptor, *Methoprene-tolerant* (*Met*), appeared around the time of segmented germ band formation and extended through dorsal closure. *Met* transcript levels declined when the JH III titer spiked. *Krüppel homolog 1* (*Kr-h1*) is a highly conserved gene that is induced by JH (*Konopova et al., 2011*; *Smykal et al., 2014*; *Belles, 2020a*; *He and Zhang, 2022*). Low levels of *Kr-h1* transcripts were present at 12 hr after egg deposition, but then were not detected until about 6 d AEL when JH-III first appeared. A spike in *Kr-h1* transcripts coincided with the JH III peak at the end of embryogenesis. Myoglianin (myo) is a member of the Transforming Growth Factor-β (TGF-β) family of morphogens (*Upadhyay et al., 2017*). Its appearance in the last larval or nymphal instar suppresses JH production (*Ishimaru et al., 2016*; *Kamsoi and Belles, 2019*; *He et al., 2020*) and is essential for the onset of metamorphosis. Myo also acts on some target tissues in the last larval instar to make them competent to initiate metamorphosis (*Awasaki et al., 2011*). *Myo* transcripts appeared in *Thermobia* after formation of the segmented germ band. They peaked during the major phase of embryonic growth, and then were low after definitive dorsal closure.

## Effects of suppression of the late embryonic peak of JH

To examine the role of embryonic JH (*Figure 2A*), we treated embryos with 7-ethoxyprecocene (7EP), a drug that suppresses JH production (*Bowers and Martinez-Pardo, 1977*; *Aboulafia-Baginsky et al., 1984*). As seen in *Figure 3A*, treatment of young juvenile *Thermobia* with 7EP markedly suppressed

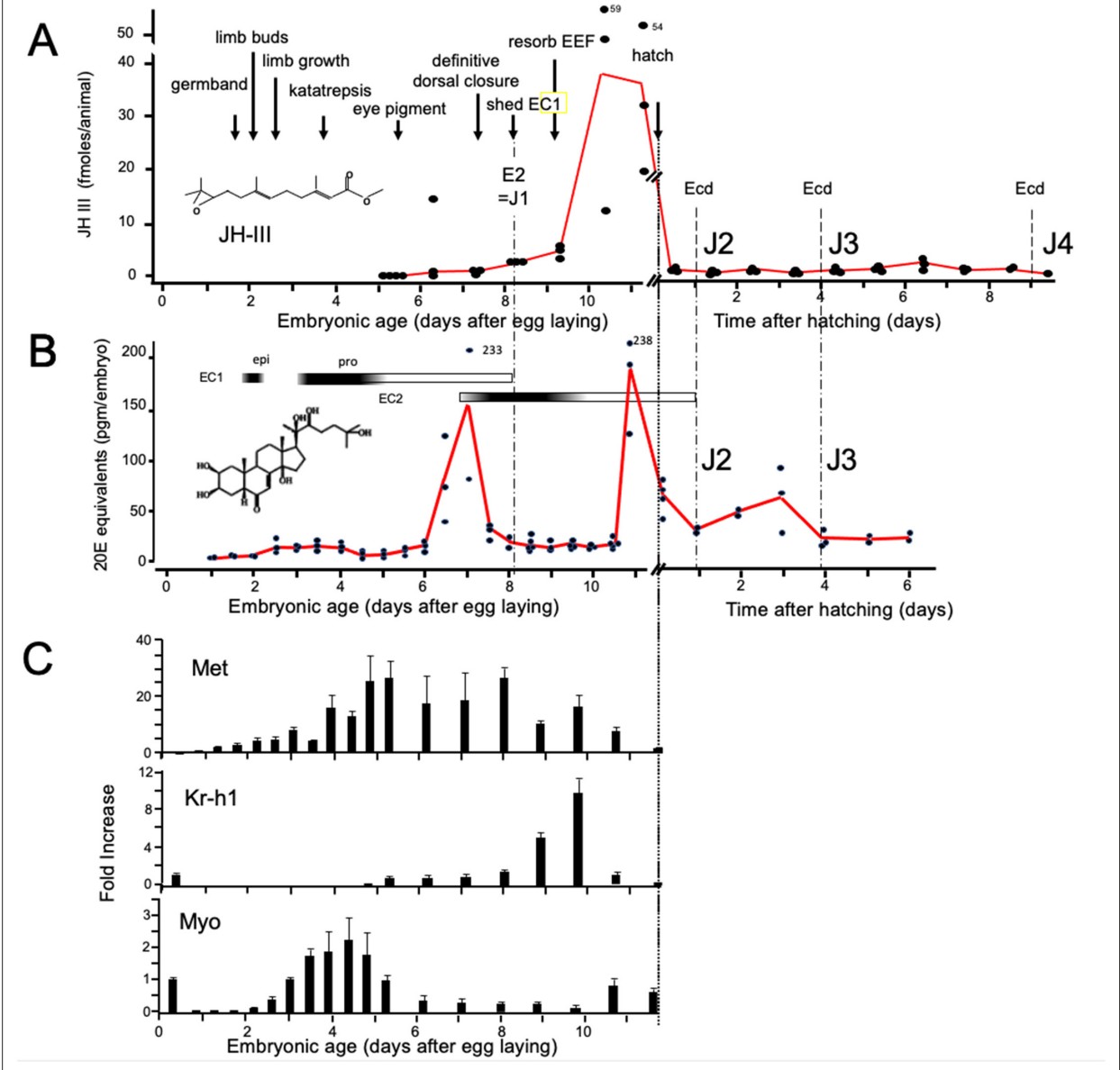

**Figure 2.** Titers of hormones and hormone related gene transcripts during embryogenesis of *Thermobia domestica*. (**A**) Titer of juvenile hormone III (JH-III) during the last 60% of embryogenesis and the first eight days of juvenile life. The timing of various milestones of embryonic development are noted for this panel and those below. ecd: ecdysis; J#: start of second, third and fourth juvenile instars. (**B**) The ecdysteroid titer in 20-hydroxyecdysone (20E) equivalents during embryogenesis and the first six days of juvenile life. Dark bars indicate the approximate time of deposition of the first embryonic (EC1) epicuticle and procuticle, and the second (EC2) embryonic cuticle (based on ***Konopová and Zrzavý, 2005***). epi: epicuticle deposition, pro: procuticle deposition. (**C**) The relative levels of transcripts of the JH receptor, *Methoprene-tolerant* (*Met*), the JH response gene *Krüppel homolog 1* (*Kr-h1*), and the TGF-β family member *Myoglianin* (*Myo*) based on real-time PCR of timed embryos. Expression of each gene is related to the 12 hr timepoint which is given the value of 1. Each bar shows the mean (+/-S.D.) for three biological replicats.

The online version of this article includes the following source data for figure 2:

**Source data 1.** Age of samples extracted for Juvenile Hormone III measurements.

**Source data 2.** Age of samples extracted for ecdysteroid measurements (given as 20 hydroxyecdysone equivalents).

their production of JH III. Also, application of 7EP to embryos at 6 d AEL, prior to the onset of JH secretion, prevented the appearance of *Kr-h1* transcripts, but the latter could then be induced by treatment with 1 ng of pyriproxyfen, a JH mimic (JHm) (***Figure 3B***). Therefore, 7EP is effective in suppressing JH synthesis in *Thermobia*.

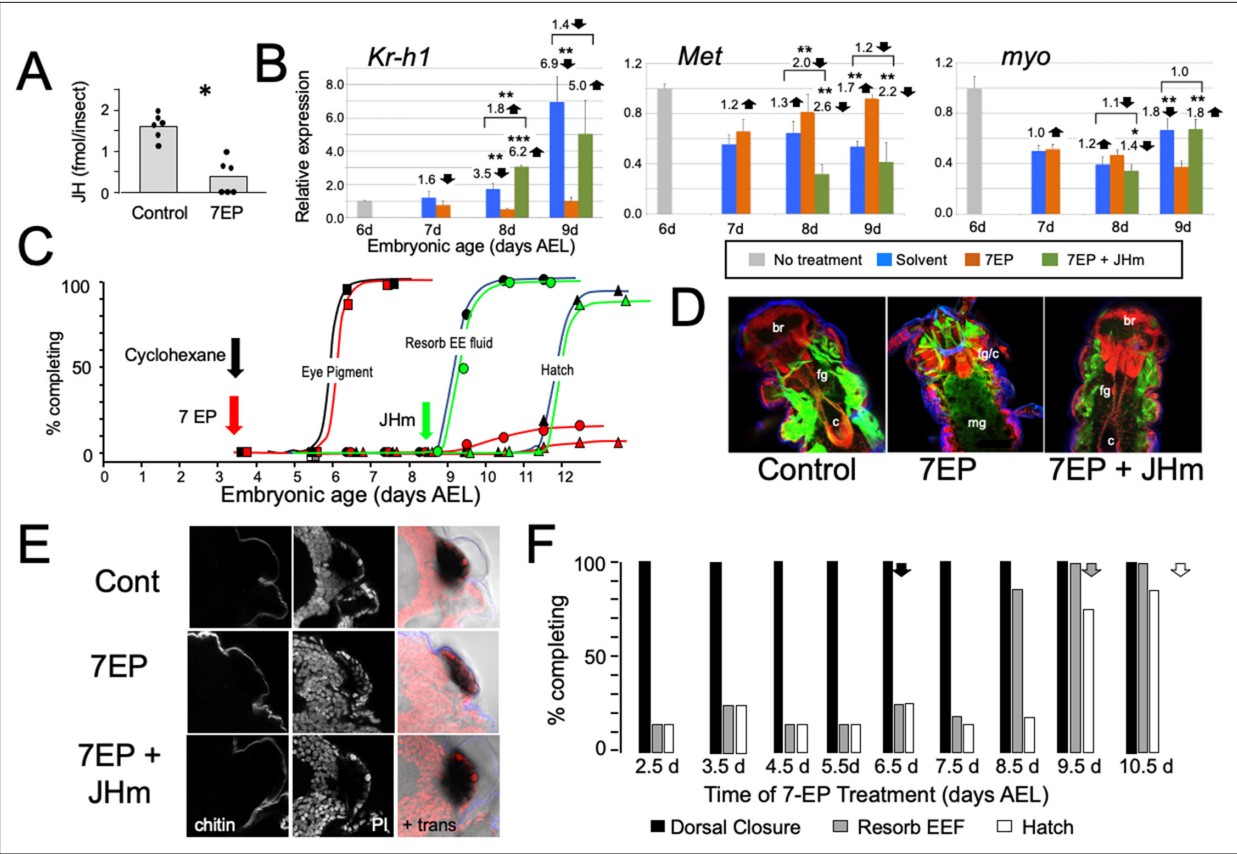

**Figure 3.** The effects of suppression of JH production on the embryonic development of *Thermobia*. (**A**) Compared with solvent alone (Control), treatment of stage 2 juveniles with 7-ethoxyprecocene (7EP) resulted in a >75% reduction in JH-III levels as measured on the day after application. (**B**) The effect treating embryos at 6 d AEL with the cyclohexane solvent (blue) or 1 µg 7EP (orange) on their subsequent expression of *Kr-h1*, *Met*, and *myo* over the next 3 days as revealed by real-time PCR. Sub-groups of 7EP treated embryos were treated with 1 ng JHm at 7 d and measured over the next 2 days (green). Significant differences determined by t-test: *=p < 0.05; **=p < 0.01; ***=p < 0.001. Numbers above columns indicate fold upregulation (upward arrow) or downregulation (downward arrow) of the two hormone treatments relative to cyclohexane control, or to the effect of JHm rescue to 7EP treated embryos. Each based on three biological replicates. (**C**) A group of 20 embryos were treated with cyclohexane (black) or 1 µg 7EP (red) and their subsequent development monitored until hatching. Both groups started eye pigmentation at the same time, but resorption of the extraembryonic fluid and hatching were suppressed in the 7EP group. The latter two events were restored by treating embryos with the 1 ng JHm at 8.5 d AEL (green). (**D**) Confocal sections of the dorsal view of embryos treated as in 'C' and examined at the time of hatching of the controls. 7EP treatment prevented extension of the foregut (fg) and crop (c) and the posterior displacement of the midgut (mg). Normal gut development was restored with JHm treatment at 8.5d. br: brain; muscle (green); propidium iodide staining (red). (**E**) Pseudo-transmitted light and confocal sections showing cuticle and nuclei of the eye region of 10 d embryos that had been treated with cyclohexane (control) or 7EP at 5 d AEL. A subset of the latter was then given JHm on day 7.5. Control and JHm-treated embryos show local apolysis of the eye cuticle and expansion of the depth of the eye due to growth of the rhabdoms; embryos treated with 7EP alone failed to show this growth. PI: propidium iodide stain, +trans: pseudo transmitted light. (**F**) The relationship of the final phenotypes of embryos to the time of their treatment with 1 µg of 7EP. Arrows indicate the normal timing of the event. EEF: extraembryonic fluid.

Initially, embryos were treated with 1 µg of 7EP at 3.5 d AEL and monitored daily to track features of development that could be viewed through the eggshell (*Figure 3C*). Solvent control and 7EP-treated embryos acquired eye pigment and underwent dorsal closure (not shown) at the same time, but the 7EP group failed to resorb their extraembryonic fluid and did not hatch. Fluid resorption and hatching was rescued when these embryos were treated with JHm shortly after dorsal closure.

Dissection of the 7EP-treated embryos showed that they usually did not shed their E1 cuticle or expand their E2 cuticle. After dorsal closure, the foregut normally lengthens as the midgut is displaced posteriorly to the anterior abdomen, but these events did not occur after the 7EP treatment (*Figure 3D*). In the case of the developing eye, the various ommatidial cell types are present and recognizable by dorsal closure and assume their mature configuration by hatching. Also, by hatching the distal most cells in each ommatidium have detached from their overlying cuticle in preparation for making the cuticular lenses of the J2 stage. Ommatidial maturation and cuticle detachment did

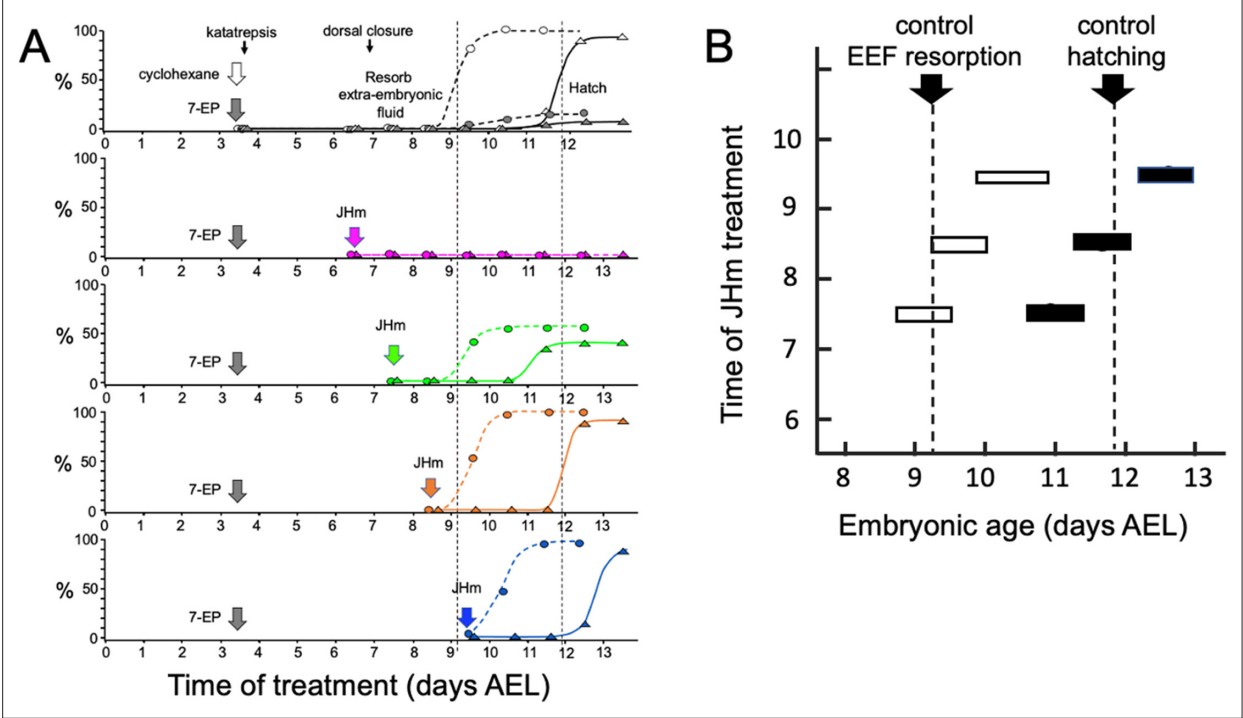

**Figure 4.** Time-course of the rescue of 7EP-imposed developmental arrest by treatment with JHm. (**A**) Groups of about 20 embryos were treated with cyclohexane (controls) or 1 µg 7EP at d3.5 AEL and then monitored daily for the time of reabsorption of the extraembryonic fluid and hatching. The vertical dashed lines indicate the 50% time for these two developmental events in the control group. Replicate groups were also given JHm (1 ng pyriproxyfen) at the indicated times and their development followed to hatching. (**B**) Summary of the timing of resorption of the extraembryonic fluid (EEF, white bars) and of hatching (black bars) for 7EP-treated embryos, whose development was rescued by JHm treatment at the indicated times.

The online version of this article includes the following source data for figure 4:

**Source data 1.** The progression of embryonic development of embryos treated with solvent or 7-ethoxyprecocene (7EP) at 3.5 days of development.

not occur after 7EP treatment (*Figure 3E*). However, the normal maturation of both the eye and the midgut was seen after subsequent application of JHm (*Figure 3D and E*).

Sensitivity to the suppression of JH synthesis by 7EP treatment did not end in an all-or-none fashion (*Figure 3F*). Development up through dorsal closure was unaffected, regardless of time of 7EP treatment, resorption of the extraembryonic fluid was suppressed by treatment up until about 8 d, and hatching was blocked until about 9 d AEL. This progressive loss of the effectiveness of suppressing JH biosynthesis suggests that JH does not provide a phasic signal, but rather acts tonically through the end of embryogenesis to support terminal differentiation and hatching.

*Figure 4A* shows that the timing of JHm application was essential for rescuing the developmental block imposed by 7EP. Embryos given 7EP on 3.5 d AEL were then treated with JHm (1 ng of pyriproxyfen) at various days thereafter, and their subsequent time of fluid resorption and hatching determined. The embryos of the group treated on 6.5 d AEL were prior to definitive dorsal closure, and all failed to complete this process. Day 7.5 AEL is about the midpoint for embryos undergoing definitive dorsal closure (*Figure 1*) and half of the treated group failed to complete dorsal closure while the remainder subsequently resorbed their extraembryonic fluid and hatched. Essentially all the 7EP-treated embryos given JHm on 8.5 d and 9.5 d AEL also showed fluid resorption and hatching. Importantly, the time of fluid resorption and hatching was time-locked to the time of JHm treatment (*Figure 4A and B*). The 7.5 d AEL group hatched a day earlier than untreated controls, the 8.5 d group hatched the same day as controls, and the 9.5 d group hatched one day later.

Manipulation of JH levels affected the production of *Kr-h1* transcripts. Suppression of JH production also blocked induction of *Kr-h1*, while application of JHm to blocked embryos reinstated elevated *Kr-h1* expression (*Figure 3C*). Transcripts for *Met*, the JH receptor, show a mild down-regulation when JH III appears late in embryogenesis (*Figure 2C*). This down-regulation was blunted by suppression of JH synthesis, but then enhanced by application of a JHm. Compared to their peak abundance at

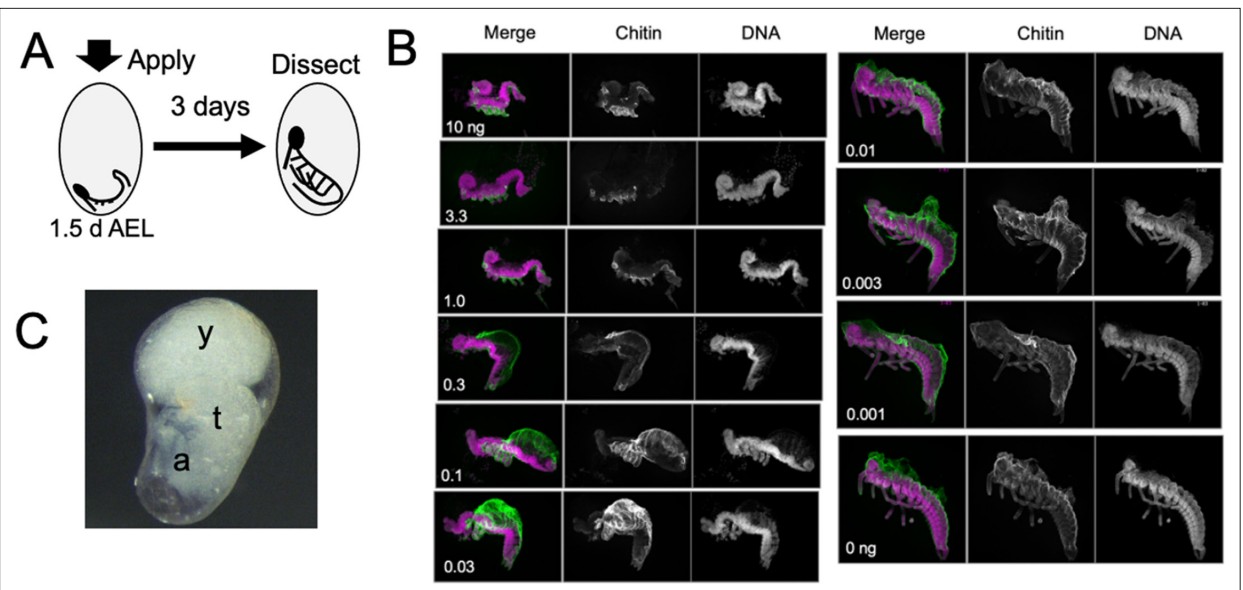

**Figure 5.** The response of *Thermobia* embryos to various doses of the JH mimic pyriproxyfen. (**A**) Cartoon showing the time of JHm treatment versus the time of dissection. (**B**) Lateral projections of confocal stacks showing the appearance of typical embryos three days after treatment with the indicated dosage of pyriproxyfen. Embryos were stained for DNA (magenta) and chitin (green). The embryos were dissected away from the yolk which caused disruption in the dorsal thoracic and head regions of some embryos. (**C**) Micrograph of an embryo arrested in mid-katatrepsis. During its envelopment of the yolk, the contraction of the amnion segregated the yolk (y) from the embryo which hangs in the ventral half of the egg. t: thorax, a: abdomen.

mid-embryogenesis, levels of *myo* transcripts were quite low going into the terminal stages of embryogenesis (*Figure 2C*). These levels were largely unaffected by late manipulations of JH (*Figure 3B*).

## Effects of early treatment with juvenile hormone

*Rohdendorf and Sehnal, 1973* first showed that treating *Thermobia* embryos with natural JHs and a wide array of synthetic analogs produced a range of embryonic deformities. We initially tested the effectiveness of JH III and its precursor, methyl farnesoate (MF), as well as three commonly used JH mimics, methoprene, hydroprene, and pyriproxyfen (*Goodman and Cusson, 2012*). When applied at 3 d AEL, all produced a full range of embryonic abnormalities, depending on dosage (data not shown). Between the naturally occurring compounds, JH III was about eightfold more potent than MF, with an $ED_{50}$ for topical application of 60 ng/egg while MF had an $ED_{50}$ of 480 ng/egg. The synthetic JH mimics were about 100 times more potent than JH III. We used pyriproxyfen (referred to as JHm) for all the following studies. The JH receptor has a higher affinity for pyriproxyfen than for JH III or methoprene (*Charles et al., 2011*; *Jindra et al., 2015*).

*Figure 5* examines the response of embryos to treatment with different doses of JHm at 1.5 d AEL, when the forming germ band is first evident at the posterior pole of the egg. Embryos were then dissected and scored at 4.5d AEL, about a day after the normal completion of katatrepsis. Embryos treated with the highest doses of JHm (1–10 ng/egg) always formed a complete segmented germ band, with its appendage buds. However, the embryos retained a S-shaped posture and stayed at the posterior pole of the egg. They remained open dorsally from the head to the end of the abdomen and subsequent growth was suppressed. The cells of the amnion did not extend to enclose the yolk and these embryos often later sunk back into the yolk. Despite their early arrest, though, they secreted a cuticle over their ventral epidermis (chitin staining in *Figure 5B*).

Embryos exposed to intermediate doses of JHm (0.3–0.03 ng pyriproxyfen) were able to subsequently 'zip up' the dorsal midline of their abdomen, which pushed the end of the abdomen ventrally and forward, causing the embryo to assume a 'comma' shape just prior to the start of katatrepsis. Subsequent morphogenesis of the head and thorax, though, was suppressed. Chitin-rich cuticle covered the abdomen and the ventral epidermis of the head and thorax. In many cases, the amnion failed to subsequently spread dorsally to enclose the yolk. Sometimes the amnion enclosed the yolk, but the lateral margins of the embryo were not pulled with it so that the yolk ended up as a rounded

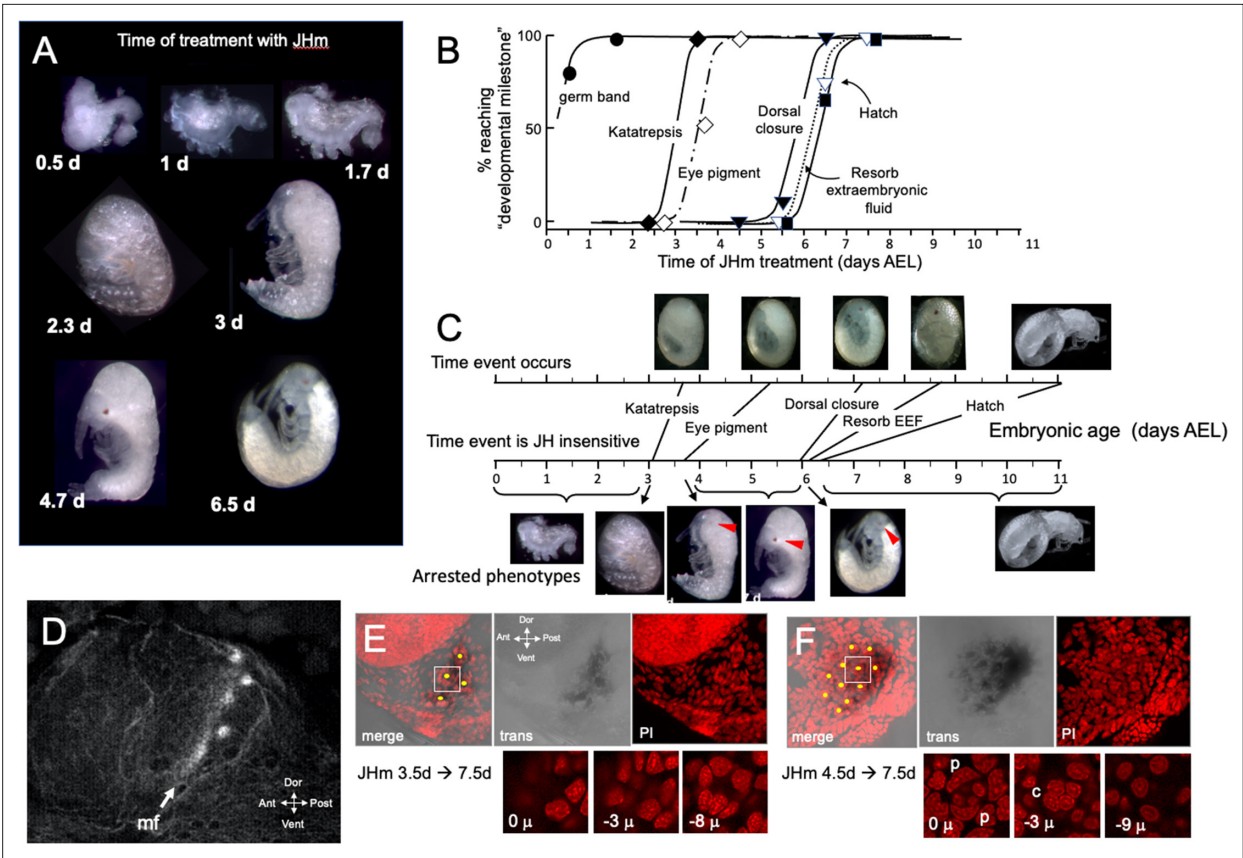

**Figure 6.** The response of *Thermobia* embryos to 1 ng JHm given at different times through embryogenesis. (**A**) Examples of the terminal phenotypes of embryos resulting from JHm treatment at the indicated times [days after egg laying (AEL)]. Yolk was dissected away from the first three embryos. The remaining embryos had undergone provisional dorsal closure with the amnion enclosing the yolk mass. (**B**) Graph showing when each developmental 'milestone' was no longer suppressed by JHm treatment. (**C**) Comparison of when developmental milestones were no longer suppressed by JHm (based on *Figure 1B*) versus their normal time of occurrence. Red triangle in micrograph shows first appearance of eye pigment. (**D**) Confocal image of actin staining showing the state of patterning of the eye primordium at katatrepsis. The morphogenetic furrow (mf) had just started and had organized only the first few posterior ommatidia. (**E, F**). Lateral confocal and pseudo transmitted light images of the eye region of 7.5 d embryos that had been treated with JHm at (**E**) 3.5 d or (**F**) 4.5 d AEL. In (**E**) eye patterning arrested early resulting in only 3–4 proto-ommatidial clusters forming in the posterior margin of the eye primordium. In (**F**) the entire eye primordium was patterned. Yellow dots identify the center of each forming ommatidium. The insets below show various Z depths of the boxed cluster in the merged image, with '0 µ' being at the surface. Nuclei of identifiable ommatidial cell types can only be recognized in (**F**). c: quartet of crystalline cone cells, p: crescent-shaped nuclei of the paired cells that secrete the cuticular lens.

The online version of this article includes the following source data and figure supplement(s) for figure 6:

**Source data 1.** The relationship of the time of treatment with a JH mimic (1 ng pyriproxyfen) to when embryonic development subsequently stalled.

**Figure supplement 1.** Confocal images showing the effects of treatment of day 1 *Schistocerca gregaria* embryos with a JHm (pyriproxyfen) versus cyclohexane (control).

dorsal lobe with the embryo hanging ventrally in a fluid-filled space (*Figure 5C*). Embryos treated with lowest doses of JHm (0.01–0.001 ng pyriproxyfen) underwent normal katatrepsis and their amnion spread to cover the yolk to establish provisional dorsal closure.

The terminal phenotypes of embryos also varied with the time of JHm treatment (*Figure 6A*). Application to eggs within 12 hr of oviposition was problematic because the cyclohexane-treated controls also showed significant lethality. With older embryos, though, treatment with the solvent alone did not interfere with subsequent development. *Figure 6B* tracks the development of groups of embryos treated with 1 ng pyriproxyfen at various days of embryogenesis and shows that the ability to complete successive 'milestones' of development depended on the time of JHm treatment. These developmental milestones lost sensitivity to JHm in the order in which they normally appeared, but the latency between loss of sensitivity and occurrence of the respective event varied (*Figure 6C*). For example, katatrepsis was blocked by JHm until shortly before the process began, but the ability to

suppress production of the eye screening pigment, by contrast, was lost 2 days before the pigment normally appeared. In *Thermobia*, as in other insects (*Friedrich, 2006*), the eye primordium makes ommatidia progressively starting at its posterior border. The eye primordium is early in this process at 3.5d AEL, with only the most posterior ommatidia being determined (*Figure 6D*). Embryos treated at this time subsequently showed screening pigment only in this posterior part of the eye associated with a few proto-ommatidial clusters (*Figure 6E*). When JH application was delayed by 1 day, the full primordium had been determined by the time of treatment and a full eye was formed with all its ommatidial clusters (*Figure 6F*). Thus, JH does not block pigment formation per se, but rather it blocks the prior determination of the cell type that will make the pigment. Once determined, these cells subsequently produce pigment regardless of the presence or absence of JH. In the production of the much larger eye of embryos of the locust, *Schistocerca gregaria*, early exposure to JHm similarly arrested the progression of the eye morphogenetic furrow. It also caused the premature maturation of the ommatidia cell types posterior to the furrow (*Figure 6—figure supplement 1*).

Exogenous JHm no longer arrested subsequent development if given after about 6 d AEL (*Figure 6B and C*). This time is just before definitive dorsal closure and subsequent appearance of

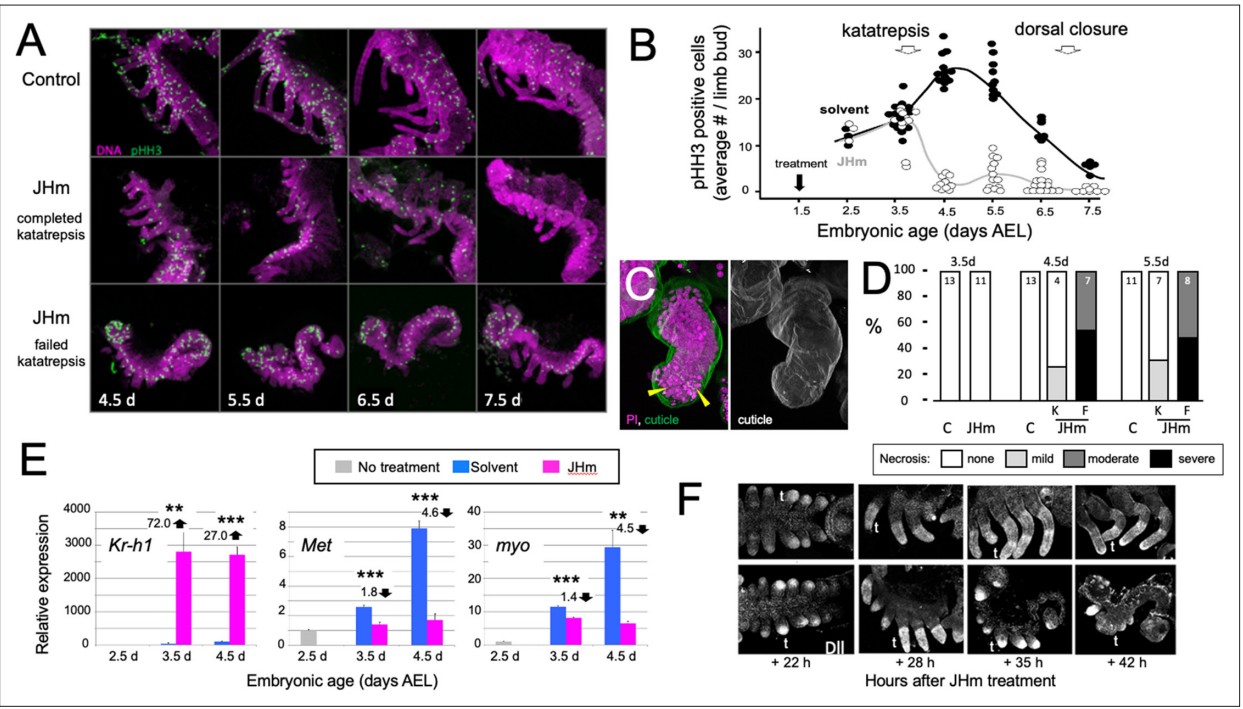

**Figure 7.** Effects of early JHm treatment on growth and patterning of the limbs. (**A**). Z-stack projections showing the lateral view of embryos that were treated with either cyclohexane (Control) or 1 ng of pyriproxyfen in cyclohexane (JHm) on d1.5 AEL and then dissected and stained on the indicated developmental days. Propidium iodide (PI) (magenta) shows nuclei and anti-phosphohistone H3 (pHH3; green) shows dividing cells. The JHm series follows a subset of embryos that did not undergo katatrepsis (bottom) and a subset that did (middle). (**B**). Summary of the number of pHH3-positive cells in the developing limb buds of JHm and solvent-treated embryos through time. Each dot records the average number of pHH3-positive cells /limb from a single embryo. (**C**). Projected Z-stack through the limb of a JHm-treated embryos that did not undergo katatrepsis. The leg produced a robust cuticle (green; gray scale image) but then started necrosis as illustrated by the highly condensed PI-positive bodies (yellow triangles) in the distal leg. (**D**). The time course of necrosis in the limbs of control and JHm-treated embryos. Necrosis was pronounced in embryos that failed katatrepsis (**F**) but rather mild in those that completed katatrepsis (**K**). (**E**). Levels of hormone-related transcripts in embryos treated at 2.5 d AEL with solvent alone (blue bars) or JHm (1 ng pyriproxyfen; pink) and then examined over the following two days. The expression is related to that on the day of treatment (2.5 d; grey) which was set as 1. *Kruppel homolog 1* (*Kr-h1*), *Methoprene tolerant* (*Met*), *myoglianin* (*myo*). Significant differences determined by t-test: *=p < 0.05; **=p < 0.01; ***=p < 0.001. Numbers above columns indicate fold upregulation (upward arrow) or downregulation (downward arrow) relative to cyclohexane control. Each based on three biological replicates. (**F**). Confocal images showing Distal-less (Dll) immunostaining at various times after treatment of embryos at 2 d AEL with solvent (**C**) or JHm. Embryos are shown from ventral view at 22 hr post-treatment, and then from lateral view thereafter. t: first thoracic leg.

The online version of this article includes the following source data for figure 7:

**Source data 1.** The effects of early treatment with a JH mimic (1 ng pyriproxyfen) on the subsequent proliferative activity in the embryonic limb buds indicated by the number of limb cells expressing phosphohistone H3 (pPH3).

endogenous JH. Shortly after katatrepsis the amnion encloses the yolk at provisional dorsal closure and these amnion cells are gradually replaced by the sides of the embryo until the latter fully enclose the yolk about 3.5 days later. JHm treatment at any time in this interval arrested further replacement. Without successful dorsal closure, the terminal events of fluid resorption and hatching did not occur. The last two events require JH but only after dorsal closure has been completed (*Figure 4*).

The effect JH on embryonic growth was assessed by treating embryos with JHm at 1.5 d AEL and then using phosphohistone H3 (pHH3; *Hendzel et al., 1997*) expression to measure the mitotic activity in limb buds (*Figure 7A and B*). At 2.5 d AEL the limb buds of control and treated embryos were of similar size and showed similar levels of pHH3 expression (*Figure 7B*). All control embryos completed katatrepsis. Their numbers of pHH3 positive cells peaked between 4.5 d and 5.5 d AEL, when their limbs had achieved their full length, and then declined. JHm-treated embryos, by contrast, varied in whether they completed katatrepsis. At 3.5 d AEL, most of the JHm-treated embryos showed numbers of pHH3 + cells like those seen in controls, although two showed a marked reduction. By 4.5 d AEL, all JHm-treated embryos showed a mitotic frequency that was only 10–15% of that of controls (*Figure 7B*). Mitosis in their limbs remained low through subsequent development.

As summarized in *Figure 7C and D*, early JHm treatment sometimes evoked necrosis in the growing limb buds. Necrotic bodies, consisting of highly compacted DNA, were rarely seen in controls at 4.5 d and 5.5 d AEL. JHm-treated embryos that completed katatrepsis also showed few, if any, necrotic bodies, but those that arrested before or during katatrepsis exhibited moderate to heavy necrosis (*Figure 7D*). In the most severe cases, the limb degenerated. Limb loss in such embryos was often stochastic, that is, in a given embryo some limbs were lost while others were maintained in a reduced state. We saw no segmental pattern as to which limbs were maintained and which were lost.

The treatment at 2.5 d AEL with JHm also affected the expression of *Kr-h1, Met*, and *myo* (*Figure 7E*). As expected from their postembryonic relationship in other insects (*Jindra et al., 2015*), exposure to JHm-induced strong *Kr-h1* expression and suppressed *Met* transcripts. For embryonic *myo*, transcript levels are highest (*Figure 2C*) during the time of peak proliferation as assessed by mitotic activity in the limb buds (*Figure 7B*). JHm treatment at 2.5d AEL resulted in a small depression of *myo* transcripts at 3.5 d AEL, but severely suppressed transcript levels by 4.5 d AEL (*Figure 7E*). These effect of early JHm treatment on *myo* expression paralleled its effect on mitotic activity in the limb bud (*Figure 7B*).

Early treatment with JHm also interrupted the proximal-distal patterning of the limb. Limb bud formation requires the expression of *distal-less* (*dll*) at the tip of the growing bud (*Cohen and Jürgens, 1989*; *Panganiban and Rubenstein, 2002*). Up through the entry into katatrepsis, the growing limb buds of JHm-treated and control embryos showed similar expression of Dll (*Figure 5F*). As leg development progresses, recruitment of additional proximal-distal patterning genes resolves Dll expression into a 'sock and band' pattern (e.g. *Angelini and Kaufman, 2005*; *Schaeper et al., 2013*) as intermediate leg regions become specified. The JHm-treated embryos in *Figure 7F* were ones that did not complete katatrepsis and they did not further refine their Dll expression. Dll immunostaining was eventually lost and their limbs degenerated. The JHm-treated embryos that completed katatrepsis, by contrast, maintained legs that were shortened but subdivided into coxa, femur, tibia, *etc.* We assume that the latter embryos had progressed far enough through their proximal-distal patterning program before the JHm-induced arrest to produce a stable leg that then maintained its integrity. An earlier arrest of the leg patterning program, however, appears to be insufficient to maintain the limb.

## The effects of JHm on cellular differentiation

While early JHm treatment suppressed growth, patterning, and cell determination, it also induced premature cellular differentiation as illustrated by forming muscles (*Figure 8*). In control embryos (*Figure 8C*), longitudinal muscle fibers become evident on 4.5 d AEL as thin bands of tissue that show weak, uniform staining for actin (as revealed by binding of fluorescent-phalloidin). The growing muscles increase in width over the next two days and their actin staining concentrates at the intersegmental boundaries where the developing muscle fibers attach to tendon cells. The appearance of myofibrils with their repeating sarcomeres becomes evident by 7.5 d AEL, but such embryos, dissected from the eggshells, showed neither spontaneous nor induced movements. By a day later, the embryos often showed weak, spontaneous movements when removed from the egg and their muscles have well-developed striations (*Figure 8C*, 8.5d). Embryos treated with JHm at 1.5 d AEL, by contrast,

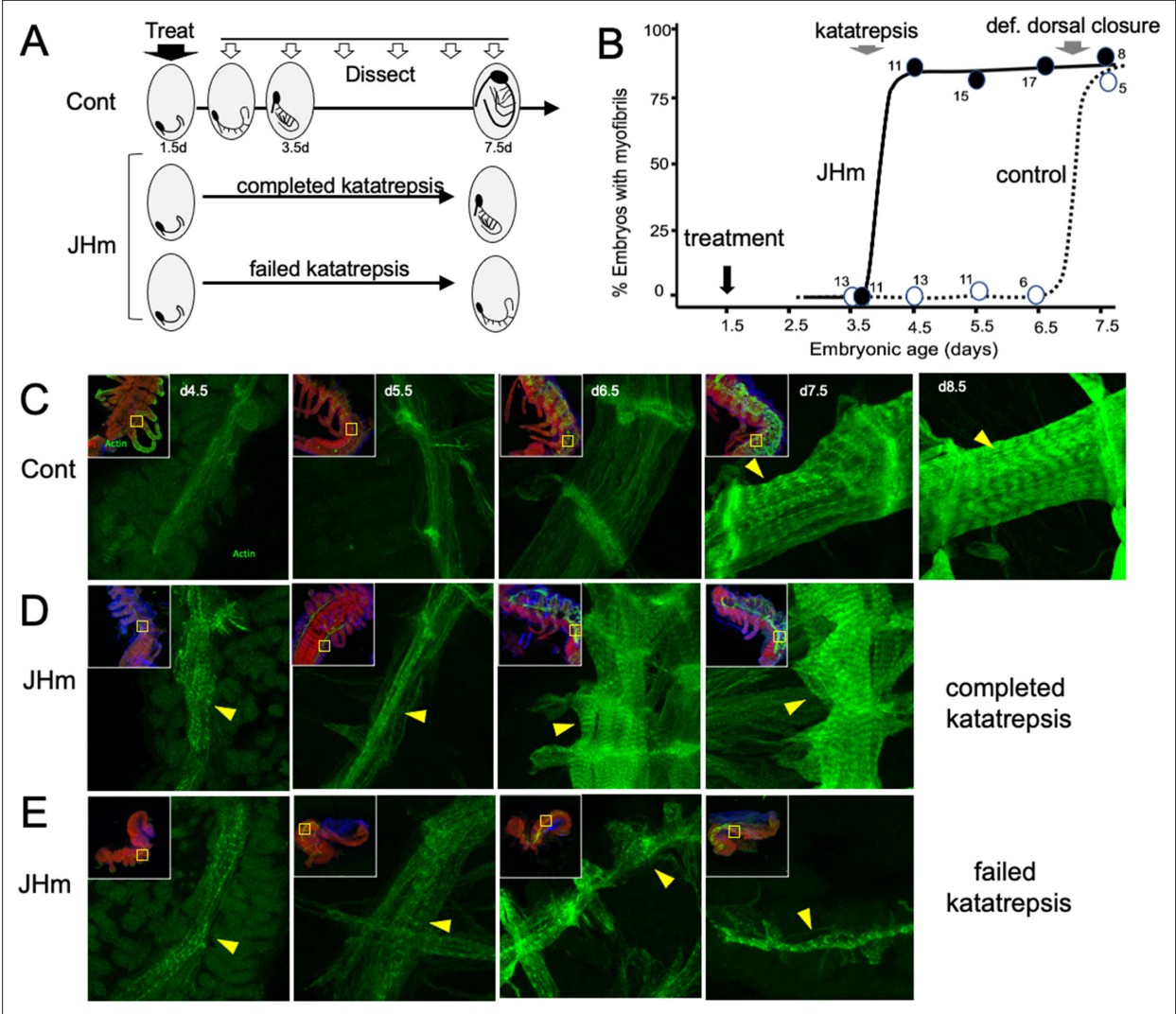

**Figure 8.** The effects of JHm in inducing early differentiation of *Thermobia* embryos. (**A**) Schematic showing the time of treatment with solvent or 1 ng pyriproxyfen (JHm) at 1.5 d AEL and the subsequent times of dissection and staining of the embryos. (**B**) Quantitation of the effects of JHm treatment on the appearance of striated myofibrils in the developing embryonic muscles. Myofibrils normally appear around the time of definitive dorsal closure, but JHm treatment at 1.5 d AEL advances their appearance by three days. Numbers are embryos examined per point. (**C–E**). Confocal optical sections showing F-actin staining in developing longitudinal muscles of control (**C**) and JHm-treated (**D, E**) embryos that were examined at the indicated days AEL. Yellow triangles indicate striations of the myofibrils. Insets show low power views of embryos with the magnified region boxed. green: F-actin shown by phalloidin binding, red: propidium iodide. JHm-treated examples are embryos that underwent katatrepsis (**D**) and ones that failed to complete katatrepsis (**E**).

showed accelerated muscle differentiation (*Figure 8D and E*). These embryos formed myofibrils by 4.5 d AEL regardless of whether or not the embryo completed katatrepsis (*Figure 8D and E*). Those that completed katatrepsis showed subsequent muscle growth, whereas those that failed katatrepsis showed little further growth.

Early JHm treatment also affected cuticle production. In *Thermobia*, the epicuticle layer of the E1 cuticle is secreted after germ band formation but prior to the start of katatrepsis (*Figure 1*; *Konopová and Zrzavý, 2005*). A fibrous procuticle is deposited after katatrepsis and covers the embryo through its period of rapid growth and dorsal closure. Chitin staining, using Calcofluor White (*Harrington and Hageage, 2003*), showed no chitin in the E1 epicuticle (as expected; *Figure 9A*, 3.5 d AEL), and only faint chitin staining of the procuticle (*Figure 9A*, 4.5–7.5 d AEL), indicating that the latter is relatively poor in chitin fibers. Strong chitin staining comes with the deposition of the E2 procuticle on 8.5 d AEL. The latter shows a lamellar ultrastructure (*Klag, 1978*; *Konopová and Zrzavý, 2005*), which

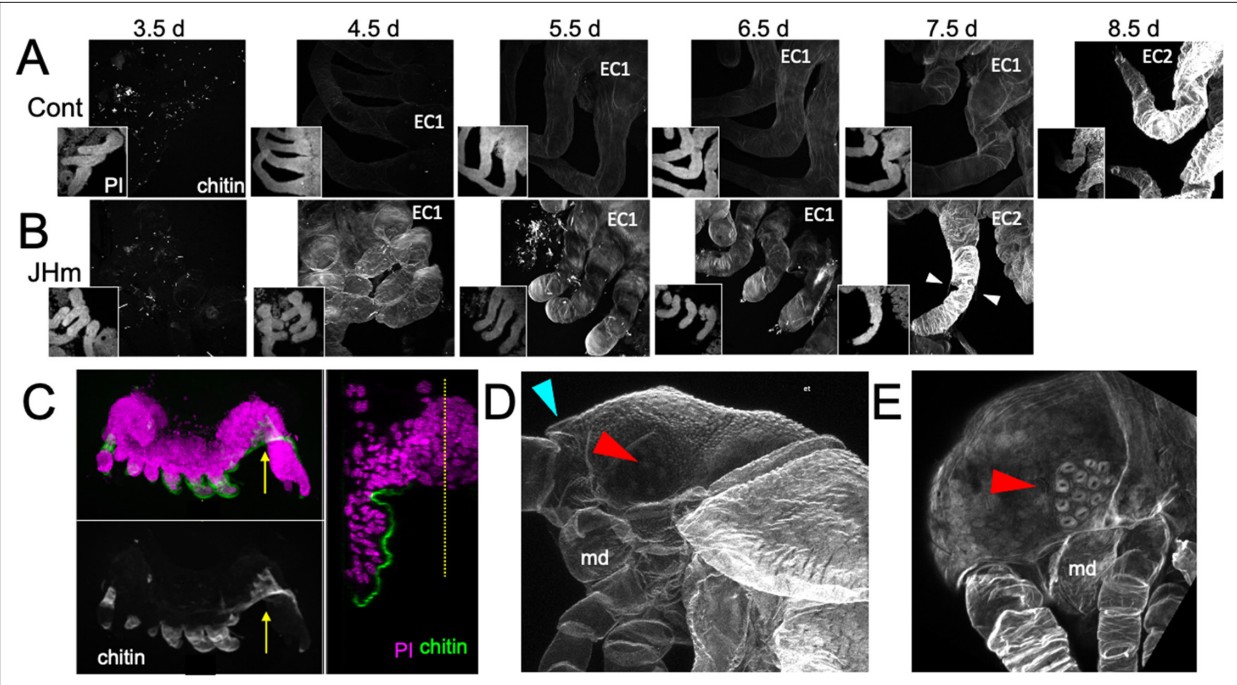

**Figure 9.** Effects of 1 ng pyriproxyfen (JHm) on the production of E1 and E2 cuticles. (**A, B**) Projected confocal Z-stacks of embryos treated with solvent (Control) or JHm at 1.5 d AEL and showing chitin staining of the developing legs at various times thereafter. The insets are propidium iodide (PI) staining of the same stack to show the form of the limbs on each day. In both groups no chitin was evident at 3.5 d AEL, when only epicuticle is present (dots are due to surface debris). In control embryos, the procuticle of the first embryonic cuticle (EC1) (starting at 4.5 d AEL) stains weakly for chitin, while the second embryonic cuticle (EC2) shows strong chitin staining (8.5d). JHm-treated embryos show enhanced chitin staining of their EC1 and they produce their EC2 a day early as demonstrated by the presence of cuticular hairs (white triangles). All cuticle images were made with the same laser power and gain. (**C**) Lateral view of a 5.5 d embryo that failed to undergo katatrepsis after JHm treatment at 1.5 d AEL. The E1 cuticle covering its ventral surface shows enhanced chitin staining. The arrow shows the plane of the transverse image on the right; the ventromedial region of the embryo is still invaginated into the yolk. PI: propidium iodide staining. Dotted line is the midline. (**D, E**) Projected confocal Z-stacks of 10 d AEL embryos showing the E2 cuticle produced by controls (**D**) or by embryos treated with JHm at 4.5 d AEL. The cuticle of the control embryo (**D**) has a pebbly surface sculpturing, an egg tooth (blue triangle) but lacks cuticular eye lenses (red triangle). The JHm-treated embryo produced a cuticle typical of later stages. It is smooth, lacks an egg tooth, and has formed thickened, but abnormal, cuticular lenses (red triangle). md: mandible.

arises from the helical deposition of chitin fibers as seen in typical insect procuticles (*Neville, 1975*; *Vincent, 1980*). Like controls, embryos treated with JHm at 1.5 d had no chitin in their E1 epicuticle (*Figure 9B*, 3.5 d AEL) but their E1 procuticle (*Figure 9B*, 4.5 d AEL) showed enhanced chitin staining. These JHm-treated embryos deposited their E2 cuticle at least a day early (*Figure 9B*, 7.5 d AEL).

The JHm-treated embryos depicted in *Figure 9B* were ones that completed katatrepsis. *Figure 9C* shows a treated embryo that failed to undergo katatrepsis, but it still secreted an E1 cuticle along its ventral surface. This cuticle also had enhanced chitin staining. The chitin-rich E1 cuticle produced in the presence of JHm is more rigid than the normal E1 cuticle. Indeed, control embryos proceeding into katatrepsis are quite delicate and easily disrupted during egg dissections, but JH-treated embryos of the same age are much more resistant to mechanical damage. As discussed below, the rigidity of the modified E1 cuticle may interfere with these embryos undergoing subsequent growth and changes in form.

JHm treatment before deposition of the E2 cuticle also changed the nature of this cuticle. As seen in *Figure 9D*, the E2 (=J1) cuticle has pebbly surface sculpturing, an egg tooth, and lacks cuticular lenses for its ommatidia. JHm treatment at 4.5 d AEL, though, redirects the epidermis to make a cuticle more like that of the J2 stage. The cuticle lacks the egg tooth and surface pebbling characteristic of the E2 stage but attempts to form J2-like cuticular lenses (*Figure 9E*). Thus, as in embryos of locusts (*Truman and Riddiford, 1999*) and crickets (*Erezyilmaz et al., 2004*), the presence of JH at the outset of the E2 molt causes the embryo to skip ahead to make cuticle characteristic of the next instar.

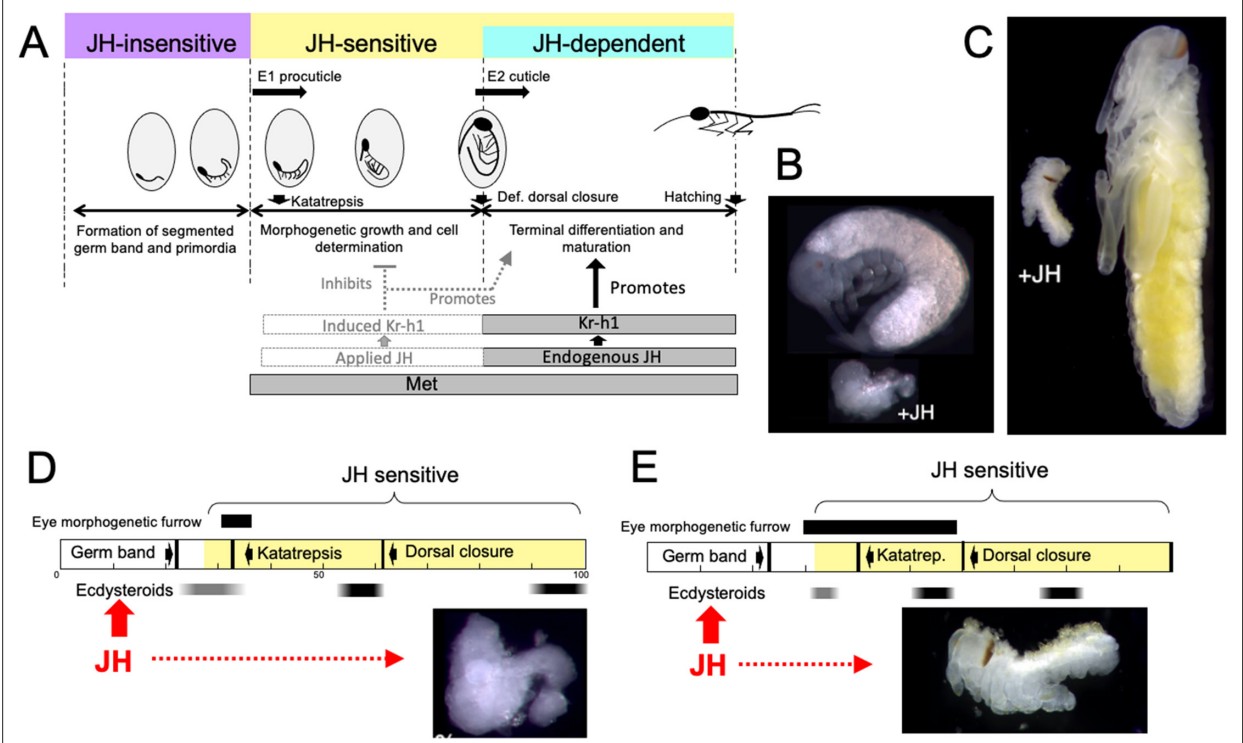

**Figure 10.** Comparison of responses of short germ band embryos to JH. (**A**) Summary of the developmental responses of *Thermobia domestica* embryos to JH. Exogenous JH has little or no developmental effect up through germ band formation. Embryonic development is then altered by JH treatment starting just before deposition of the first embryonic (**E1**) cuticle, but endogenous JH does not appear until around definitive dorsal closure, when it is needed for terminal differentiation. (**B, C**) Photomicrographs of embryos of *Thermobia* (**B**) and of *Schistocerca gregaria* (**C**) comparing the size and morphology of a control embryo at definitive dorsal closure with a clutch mate that had been treated with JHm the day after oviposition (+JH). (**D, E**) Schematic summary of embryonic development of *Thermobia* (**D**) and *Schistocerca* (**E**) comparing the time of ecdysteroid surges and the onset of JH sensitivity with the time course of embryogenesis. The black bar shows the time of progressive patterning of the eye primordium. Eye patterning begins **before** *Schistocerca* embryos become sensitive to JH, but **after** *Thermobia* embryos become JH sensitive. Consequently, treatment with JHm (red) a day after oviposition results in *Schistocerca* embryos with tiny eyes but *Thermobia* embryos lack eyes all together. The times of ecdysteroids in the locust are based on data from *Locusta migratoria* (**Lagueux et al., 1979**).

Overall, then, embryonic development of *Thermobia* becomes sensitive to exogenous JH around the time that the E1 cuticle is being produced. JH treatment suppresses subsequent morphogenesis and cell determination but promotes developmental programs characteristic of the late phases of embryogenesis.

## Discussion
### The role of JH in embryogenesis of *Thermobia domestica*

JH appears in *Thermobia* embryos after definitive dorsal closure (*Figure 2A*). At this point the embryo has completely enclosed the yolk mass and its final size is set. Morphogenetic growth is completed, the E2 cuticle that will be covering the hatchling is being induced, and developmental programs have shifted to tissue differentiation and terminal maturation. These latter processes are JH dependent (*Figure 10A*) and do not occur if JH is absent (*Figures 3 and 4*).

While loss-of-function experiments, which remove JH, produce one type of abnormal development, gain-of-function experiments, involving the providing of JH when it should be absent, produce a different, but complementary, range of abnormalities. As originally reported by *Rohdendorf and Sehnal, 1973*, JHm treatment of *Thermobia* embryos results in a diversity of arrested phenotypes that ranged down to minute embryos that lacked limbs and eyes (as in *Figure 10B*). The alteration of development by JHm treatment was first evident when the embryo was producing its E1 procuticle. For the developing limb bud, premature exposure to JHm inhibited cell division (*Figure 7A and*

*B*) and arrested proximal-distal patterning (*Figure 7F*). If arrest was early enough in the patterning process, the limb bud was not stable and began degeneration despite having deposited a modified E1 cuticle (*Figure 7C*). The effects of premature JH exposure were well illustrated for the eye (*Figure 3E*). JH had to be absent during the time when the morphogenetic furrow was moving over the eye primordium (*Figure 6D–F*). Premature exposure to JHm stopped furrow movement; the cells behind the furrow maintained their determination and made ommatidia cell types, but those in front of the furrow remained undetermined. As summarized in *Figure 6C*, then, many tissues and cell types are determined during the period from the segmented germ band to definitive dorsal closure, and exposure to JHm during this period generated a diversity of phenotypes that reflected the state of embryogenesis when the JHm was applied. Some tissues, like the eye could survive an arrest in early patterning and simply made a small, partial eye (*Figure 6E*). Other tissues like the leg bud, though, were apparently unstable if patterning was arrested too early and subsequently degenerated.

Importantly, premature exposure to JH does not just simply arrest development. Rather, it shifts cells into differentiative programs characteristic of late embryogenesis. Such a shift to later developmental programs by JHm is evident in *Thermobia* in the precocious appearance of myofibrils in developing muscle (*Figure 8D and E*), and in the production of late cuticle types by the epidermis (*Figure 9B*). Similarly, for eye development in *Schistocerca*, early treatment of embryos with JHm not only arrests the subsequent progression of the morphogenetic furrow but causes the ommatidia in the wake of the furrow to undergo premature pigment deposition (*Figure 6—figure supplement 1*).

As summarized in *Figure 10A*, *Thermobia* embryos show three distinct phases with regards to JH. The initial JH-insensitive phase extends through the formation of the segmented germ band and limb buds. Then follows a JH-sensitive phase when the embryo is responsive to treatment with JH or its mimics. The onset of JH sensitivity correlates with the appearance of Met, and the application of JHm during this period induces transcription of *Kr-h1* (*Figure 7E*), the major JH response gene (*Jindra et al., 2015*). The JH-dependent phase then corresponds with the appearance of endogenous JH around definitive dorsal closure. Morphogenetic growth is finished, and JH then supports the terminal differentiation of the determined tissues.

A systemic signal like JH that can switch the embryo from morphogenesis to terminal differentiation may be especially important for short germ band embryos. Eggs may vary in the amount of yolk that they possess (*Fox and Czesak, 2000*; *Yanagi and Tuda, 2012*). In such cases the embryo's maximum size is established once the yolk has been fully internalized at definitive dorsal closure. A systemic signal, like JH, when dorsal closure has occurred might serve to coordinate a shift to terminal differentiation programs across the embryo.

The events responsible for the appearance of Met and of JH sensitivity in the segmented germ band are not known. The fact that the first JH-induced change seen in *Thermobia* embryos is the modification of the E1 molt is interesting considering the intimate relationship between JH and ecdysteroids during the postembryonic molts of hemi- and holometabolous insects (e.g. *Riddiford, 1976*; *Riddiford, 2020*). JH acts postembryonically to enforce *status-quo* molts, but there is no *status-quo* to maintain for the first embryonic molt. While the appearance of JH sensitivity coincides with the appearance of *Met* transcripts, the question is still why should the system be sensitive to JH days before the hormone appears? One possibility is that a component of the ecdysone response system is inherently sensitive to JH. An intriguing candidate for such a protein is Taiman. The *taiman* gene encodes a steroid hormone co-activator that mediates ecdysteroid activation (*Montell, 2001*) and also serves as heterodimeric partner with Met to form the active JH receptor (*Jindra et al., 2015*; *Charles et al., 2011*). C-terminal exons of the *taiman* transcript are alternately spliced to form several isoforms in the mosquito, *Aedes aegypti* (*Liu et al., 2018*), the flour beetle, *Tribolium castaneum*, and the cockroach, *Blattella germanica* (*Lozano et al., 2014*). JH has been shown to control this splicing in both *Aedes* (*Liu et al., 2018*) and *Blattella* (*Lozano et al., 2014*). Interestingly, in mosquito reproduction, the *taiman* isoforms made in the absence of JH poorly support ecdysteroid-induced activation, while those made in the presence of JH mediate strong activation (*Liu et al., 2018*). We speculate that if a similar relationship were to hold in *Thermobia*, then *taiman* isoforms made in the JH-free environment of the early embryo might support a rather weak ecdysteroid response resulting in the thin, extensible E1 cuticle. JHm exposure, though, would change the *taiman* isoforms perhaps supporting the production of a robust cuticle, more typical of later embryonic molts.

Although early JH exposure affects embryonic growth, tissue patterning, and cuticle types, we do not know how these effects are interrelated. Growth and morphogen signaling interact in complex ways during embryogenesis, with morphogen gradients directing growth, and growth altering morphogen landscapes (*Schwank and Basler, 2010*; *Dekanty and Milán, 2011*). Does JH suppress morphogen signaling and this, in turn, suppresses growth or, alternatively, does JH suppress growth which then arrests the progression of morphogen signaling? We find that JH suppresses transcript levels of the TGF-β family member, myoglianin (*Figure 7E*). In last stage *Drosophila* larvae, this circulating morphogen matches imaginal disc growth with that of the overall animal (*Upadhyay et al., 2020*). We do not know how it functions in embryos, but it might serve a similar role in short germ band embryos to ensure that limb size is matched to overall embryo size. Whether JH suppression of myoglianin production inhibits embryonic growth or *vice versa* is unknown.

The fact that the rapidly growing embryo is covered by a cuticle presents another complication. The fibrous nature of the E1 cuticle (*Konopová and Zrzavý, 2005*) likely allows it to stretch to accommodate epidermal proliferation and limb extension. The more robust, E1 cuticle produced after JH treatment (*Figure 9B*) resists extension and this inextensibility may mechanically hinder subsequent proliferation and growth of the JH-treated embryos.

JH also interferes with the morphogenetic movements of katatrepsis. These movements can be blocked by JH until shortly before they begin (*Figure 6C*) and we also find embryos that are blocked in mid-katatrepsis. JH could be inhibiting the contraction of the serosa or its modification of the E1 cuticle may prevent the embryos from passively responding to serosa contractions. We have observed cases in which the extraembryonic membranes have contracted but the miniature embryo has been pulled into the yolk rather than around the outside of the yolk mass (as in *Figure 5C*). Such cases suggest that a loss of embryo extensibility may be a major factor in their failure to undergo katatrepsis.

## Comparative aspects of JH effects on insect embryogenesis

Embryonic JH titers have been measured or estimated for a variety of hemimetabolous and holometabolous insects (see *Belles, 2020b*). They have been directly determined for embryos of the cockroaches, *Nauphoeta cinerea* (*Imboden et al., 1978*) and *Blattella germanica* (*Maestro et al., 2010*), the locust, *Locusta migratoria* (*Temin et al., 1986*), and the moth, *Manduca sexta*, (*Bergot et al., 1981a*). Alternatively, the abundance of *Kr-h1* transcripts through embryogenesis has been used as a proxy for JH for embryos of the hemipterans *Planococcus kraunhiae,* (*Vea et al., 2016*) and *Pyrrhocoris apterus* (*Konopova et al., 2011*), the thrip, *Frankliniella occidentalis* (*Minakuchi et al., 2011*), the moth *Bombyx mori* (*Daimon et al., 2015*), and the fruit fly, *Drosophila melanogaster* (*Beck et al., 2004*). All these insects show an embryonic JH titer like that seen in *Thermobia*: JH appears around definitive dorsal closure when the final size of the embryo has been established.

Consistent with its late appearance, the inhibition of JH production suppresses late embryonic development. Late arrests were seen in *Thermobia* (*Figure 3*), the stick insect, *Clitumnus extradentatus* (*Cavallin and Fournier, 1981*), the grasshopper, *Locusta migratoria* (*Aboulafia-Baginsky et al., 1984*), the cockroach *Nauphoeta cinerea* (*Brüning et al., 1985*) and the milkweed bug, *Oncopeltus fasciatus* (*Dorn, 1982*). In the latter two cases, late development was restored by treatment with JH or JH mimics. The use of maternal RNAi treatments to knock-down transcripts of components of the JH synthesis and response pathways in *Blattella germanica* (*Fernandez-Nicolas and Belles, 2017*) also interfered with late embryonic development, although some early embryonic anomalies were also noted. The latter may be related to low levels of maternal transcripts that are loaded into the egg.

Amongst embryos from holometabolous orders, interference with JH production in the beetle *Tribolium castaneum* (*Naruse et al., 2020*) also resulted in abnormal late development. In *Bombyx*, by contrast, the genetic suppression of JH production caused only a mild delay in terminal embryogenesis but suppressed hatching. When treated larvae are freed from their shells, though, they feed and progressed to the third instar where they then attempted metamorphosis (*Daimon et al., 2015*). In *Drosophila*, the suppression of JH production had little effect on embryogenesis except causing abnormal migration of the primordial germ cells (*Barton et al., 2024*). Overall, then, although JH appears during late embryogenesis in all insects, the importance of that JH for the completion of embryogenesis wanes in the holometabolous orders.

Embryos also vary in how they respond to early treatment with JH. The most severely affected embryos are those that undergo short germband development, such as embryos of *Thermobia*

(*Rohdendorf and Sehnal, 1973*; this paper), the locusts, *Schistocerca* and *Locusta* (*Novák, 1969*; *Truman and Riddiford, 1999*; *Truman and Riddiford, 2002*) and the cricket *Acheta* (*Erezyilmaz et al., 2004*). *Figure 10B and C* compares the morphology of normal *Thermobia* and *Schistocerca* embryos at dorsal closure with clutch-mates that were treated with JHm within a day after fertilization. In both cases katatrepsis was blocked and growth was severely retarded. The most severely affected *Thermobia* embryos lacked eyes and their limbs regressed, but the most affected *Schistocerca* embryos possessed small eyes and had stable, although deformed, limbs. We think that these species differences in the severity of JH suppression of embryogenesis are due to changes in the relationship of the embryonic timetable to the timing of ecdysteroid pulses that drive embryonic molts.

The impact of changing the time of the first embryonic molt relative to the embryogenesis timetable is illustrated for eye development in *Figure 10D and E*. In *Thermobia*, the eye primordium is patterned between about 30 to 45% of development as the morphogenetic furrow moves across it. JH treatment during that period arrests the furrow, thereby determining the size of the eye that subsequently forms (*Figure 6D and E*). The embryo, though, becomes sensitive to JH suppression when it is depositing the E1 cuticle. Since cuticle deposition begins before furrow movement, the early JH treatments can prevent furrow initiation and eye development altogether. Eye patterning in locusts, by contrast, is initiated at a similar developmental time but extends until dorsal closure to make the large eye of the grasshopper nymph (*Figure 6—figure supplement 1*). Ecdysteroids and production of the E1 cuticle occurs relatively later in locusts (*Lagueux et al., 1979*), though, and the earliest rows of ommatidia have already been determined before the start of the E1 molt and the embryo becoming sensitive to JH. Consequently, JH-treated locust embryos always form an eye, although the size of that eye becomes progressively larger as JHm treatment is delayed. The difference in the extent of limb suppression in the two species also relates to how far development has progressed at the time that the embryos become responsive to JH: in *Thermobia* the legs are at an earlier stage of proximal-distal patterning as compared to those of *Schistocerca* (Truman and Riddiford, unpublished) and *Acheta* (*Erezyilmaz et al., 2004*). With the earliest JH treatments, the limbs of *Thermobia* arrest too early to maintain a stable limb, whereas in *Schistocerca* and *Acheta* embryos, arrest occurs after the establishment of the leg segments of femur, tibia, *etc*. Although stunted and deformed, these JHm-exposed legs are stable.

JH exposure at mid-stage development alters the nature of the E2 cuticle deposited by embryos of *Thermobia* (*Figure 9*), *Locusta* (*Truman and Riddiford, 1999*) and *Acheta* (*Erezyilmaz et al., 2004*). In locusts and crickets, the induced cuticle has the features of the first nymphal (=E3) cuticle, rather than bearing the hatching specializations of the E2 (=Pronymphal) cuticle. Similarly, in *Thermobia*, the E2 (=J1) cuticle produced under the influence of JH has the features of the J2 cuticle and lacks the egg tooth and surface sculpturing that are features of the E2 cuticle (*Figure 9D and E*). The similarity in the effects of JH on E2 cuticle production in *Thermobia* and in polyneopteron hemimetabolous insects (locusts and crickets) suggests that their last common ancestor would have responded to JH exposure in a similar way. Hence, the ancestral effects of JH on molting were in evoking *progressive* molts to the final juvenile condition. The function of JH in maintaining *status quo* molts came later, when this hormone began being used during postembryonic molts.

Insects that show intermediate germ band development such as the linden bug, *Pyrrhocoris apterus,* devote more of the blastoderm to the production of the germband than do the short germband forms (*Anderson, 1972a*). Premature exposure to JH also suppresses katatrepsis and dorsal closure but their growth suppression is not as severe as seen in *Thermobia* and locusts (*Enslee and Riddiford, 1977*), likely because the embryos are relatively larger when they become responsive to JH. JHm treatment also causes an advancement in embryo pigmentation in *Pyrrhocoris* (*Enslee and Riddiford, 1977*).

Embryos of holometabolous insects typically show either intermediate or long germband development (*Anderson, 1972b*) and they are the least affected by exposure to exogenous JH. These embryos form their germband in its 'appropriate' position with its head directed to the anterior pole of the egg and, hence, does not show the movements of katatrepsis (*Anderson, 1972b*). The elongated embryos of the Lepidoptera, though, show an unrelated set of movements around dorsal closure (*Heming, 2003*; *Anderson, 1972b*) to reposition the embryo so that its dorsal surface is against the shell. Lepidopteran embryos enclose only part of the yolk at dorsal closure and subsequently consume the remainder of the yolk before hatching. JH treatment to embryos of the giant

silk moth, *Hyalophora cecropia,* and the sphinx moth, *Manduca sexta* (*Riddiford and Williams, 1967*; L.M.Riddiford, unpublished) blocks the repositioning of the embryo and its ventral surface remains against the shell. Although there is physical distortion caused by their ventral-out position, these embryos finish their development in a relatively normal manner. These embryos, though, can neither consume the extraembryonic yolk nor hatch.

Long germband embryos, as seen in Diptera and Hymenoptera, are least affected by JHm treatment. As best characterized in *Drosophila* (*Cohen, 1993*), almost all the blastoderm is devoted to making the germband, and the body axis is established simultaneously along its length, with organ primordia established soon thereafter. Compared to their development in short germband embryos, the embryonic primordia for structures like the eyes (*Friedrich, 2006*) and legs (*Tanaka and Truman, 2007*) only undergo early phases of their patterning, with the patterned portion forming the larval structure, and the remainder set aside as a dormant primordium. In the epidermis of *Drosophila,* most of these persisting embryonic primordia are sequestered as invaginated imaginal discs and contribute nothing to the larval body. In species with less derived development, the imaginal primordia are embedded within the larval epidermis and help in making the larval structure. Compared to their short germband relatives, the primordia of long germband embryos may show modest or little growth to establish the appendage of the hatchling. This de-emphasis on proliferative growth removes the feature that was most sensitive to exogenous JH suppression in short germband embryos. Of course, as covered in the next section, this phase of proliferative morphogenesis has been translocated from the embryo to the last larval instar in holometabolous insects and its JH sensitivity may have been carried with it.

## Relevance of the embryonic effects of JH in *Thermobia* to its postembryonic effects in hemimetabolous and holometabolous insects

Although insects show similar titers of JH during embryogenesis, they differ in their pattern of JH production during postembryonic life (*Figure 11A*). The only information for ametabolous insects is for *Thermobia domestica*. Direct hormone measurements (*Figure 2A*), and *Kr-h1* transcript levels (*Fernandez-Nicolas et al., 2023*) indicate that JH levels are low at hatching and remain low throughout juvenile growth and into the adult. In hemimetabolous insects, by contrast, JH titer measurements or *Kr-h1* expression data are available for examples of paleopteran (mayflies; *Kamsoi et al., 2021*), polyneopteran (including grasshoppers and cockroaches: *Temin et al., 1986*; *Lanzrein et al., 1985*; *Maestro et al., 2010*) and paraneopteran orders (heteropteran, *Konopova et al., 2011*). High levels of JH are present through the penultimate nymphal instar but then disappear in the last nymphal stage. This decline in JH is both necessary and sufficient for metamorphosis to the adult. In the Holometabola, JH levels are high through the penultimate larval stage but then decline early in the last larval stage, as the larva becomes committed to metamorphosis. After pupal commitment, JH transiently returns during the prepupal period. This reappearance of JH is needed by some tissues to prevent their premature 'jump' to the adult stage (*Williams, 1961*). JH must then be removed from the system to allow the transition of the pupa into the adult.

As discussed above, the similarity of the effects of JH in embryos of *Thermobia* and of polyneopteran hemimetabolous insects argue that JH had the ability to suppress morphogenesis and promote terminal differentiation in the last common ancestor before these groups diverged about 400 mya. How, then, do these ancestral, embryonic actions of JH relate to its modern function of regulating metamorphosis, and what were the selective pressures that moved JH into the postembryonic realm?

The embryonic actions of JH are compared with its postembryonic actions in *Figure 12*. In the ancestral condition, JH can act in the embryo to suppress morphogenesis and promote differentiation, but, since it normally appears after morphogenesis and tissue patterning are complete, its main embryonic function is to support terminal differentiation. In *Thermobia,* JH levels are then quite low in the juvenile (*Figure 2A*; *Fernandez-Nicolas et al., 2023*), indicating that JH is not needed to maintain the juvenile form. *Watson, 1967* showed that the appearance of scales in its J4 stage is suppressed by JH application, but we have not been able to induce premature scale formation in the J3 stage by treatment of the J1 and J2 stages with 7EP (Truman and Riddiford, unpublished). We do not think that the appearance of scales is *caused* by a modulation in the low postembryonic JH levels. Rather, the ability of JH to suppress scale morphogenesis in the juvenile is likely an extension of its morphostatic actions that were evident in the embryo.

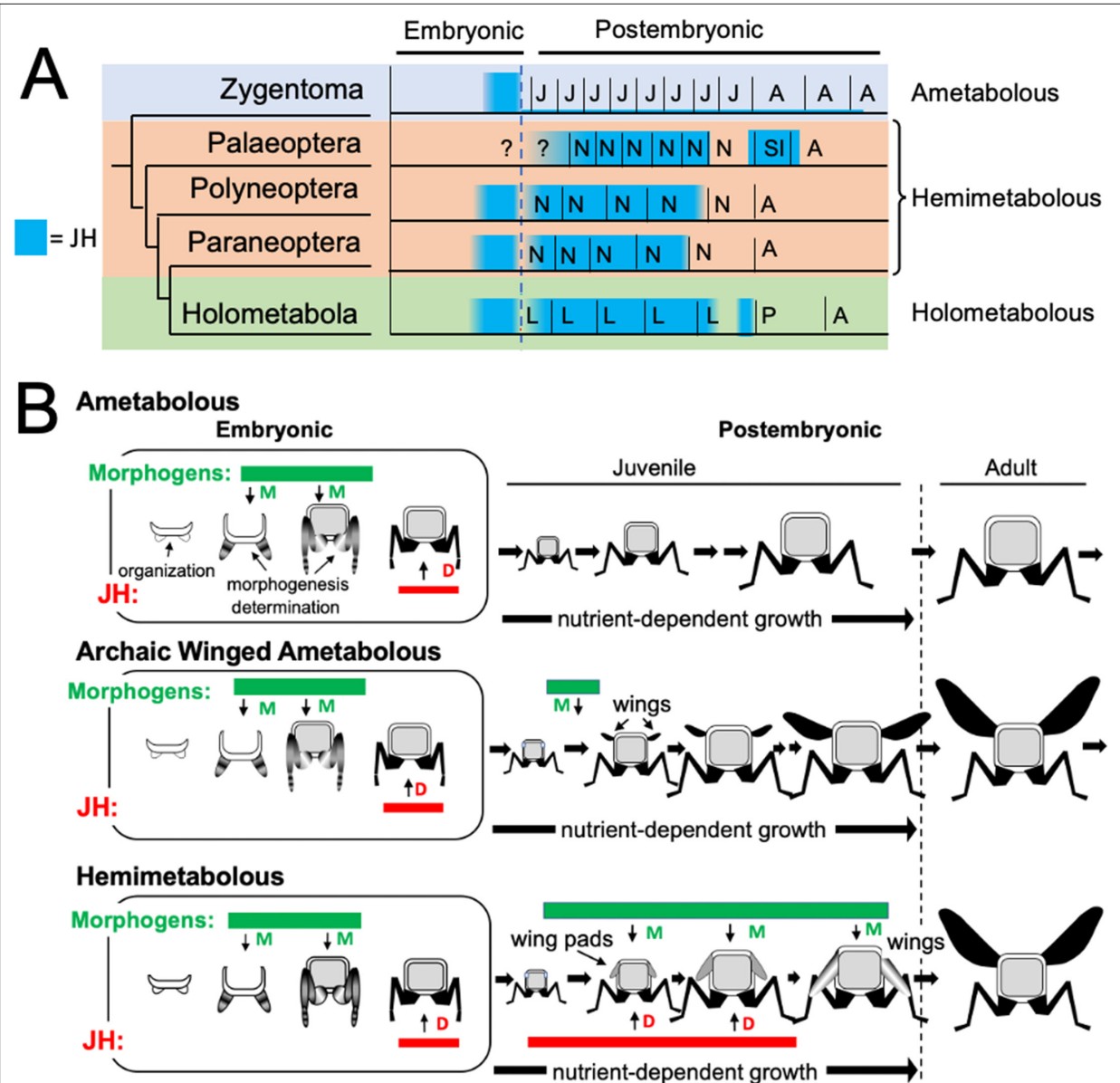

**Figure 11.** Phylogenetic shifts in JH production during embryonic and postembryonic development in insects. (**A**) Depiction of JH titers in ametabolous, hemimetabolous, and holometabolous insects. Titers are based on direct measurements or are estimated from *Kr-h1* expression. The ametabolous order, Zygentoma, provides the only data for an ametabolous insect, *Thermobia* (*Figure 2A*; *Fernandez-Nicolas et al., 2023*). Palaeopteran postembryonic data from the mayfly, *Cloeon dipterum* (*Kamsoi et al., 2021*). Polyneopteran titers based on locusts (*Temin et al., 1986*; *Truman and Riddiford, 1999*) and *Blattella* (*Maestro et al., 2010*). Paraneopteran from *Pyrrhocoris* (*Konopova et al., 2011*) and Holometabolous for *Manduca* (*Bergot et al., 1981a*; *Borst et al., 1987*; *Fain and Riddiford, 1975*). (**B**) A scenario for the role of postembryonic JH in the evolution of the wing pad. In the ametabolous condition, as typified by *Thermobia*, the major phase of body morphogenesis (M) is confined to mid-embryogenesis, followed by the appearance of JH which supports differentiation (D). In archaic winged insects a new, postembryonic phase of morphogenesis supports wing formation in the young juvenile. The small wings then undergo positive allometric growth until they are large enough to support flight of the older juveniles and the adults. The postembryonic, reappearance of JH during wing morphogenesis redirected development to make the wing pad. This compromise developmental program is then maintained until the end of juvenile growth when the disappearance of JH allows wing differentiation.

Unlike the juveniles of *Thermobia*, the nymphs and larvae of hemi- and holometabolous insects have long been known to need JH to maintain their respective states. However, more recent studies involving the loss of JH or JH function in the silkworm, *Bombyx mori* (*Daimon et al., 2015*), and the bug *Pyrrhocoris apterus* (*Smykal et al., 2014*) show that the early juvenile instars are maintained regardless of the presence or absence of JH or its main effector, Kr-h1. The juveniles shift from a

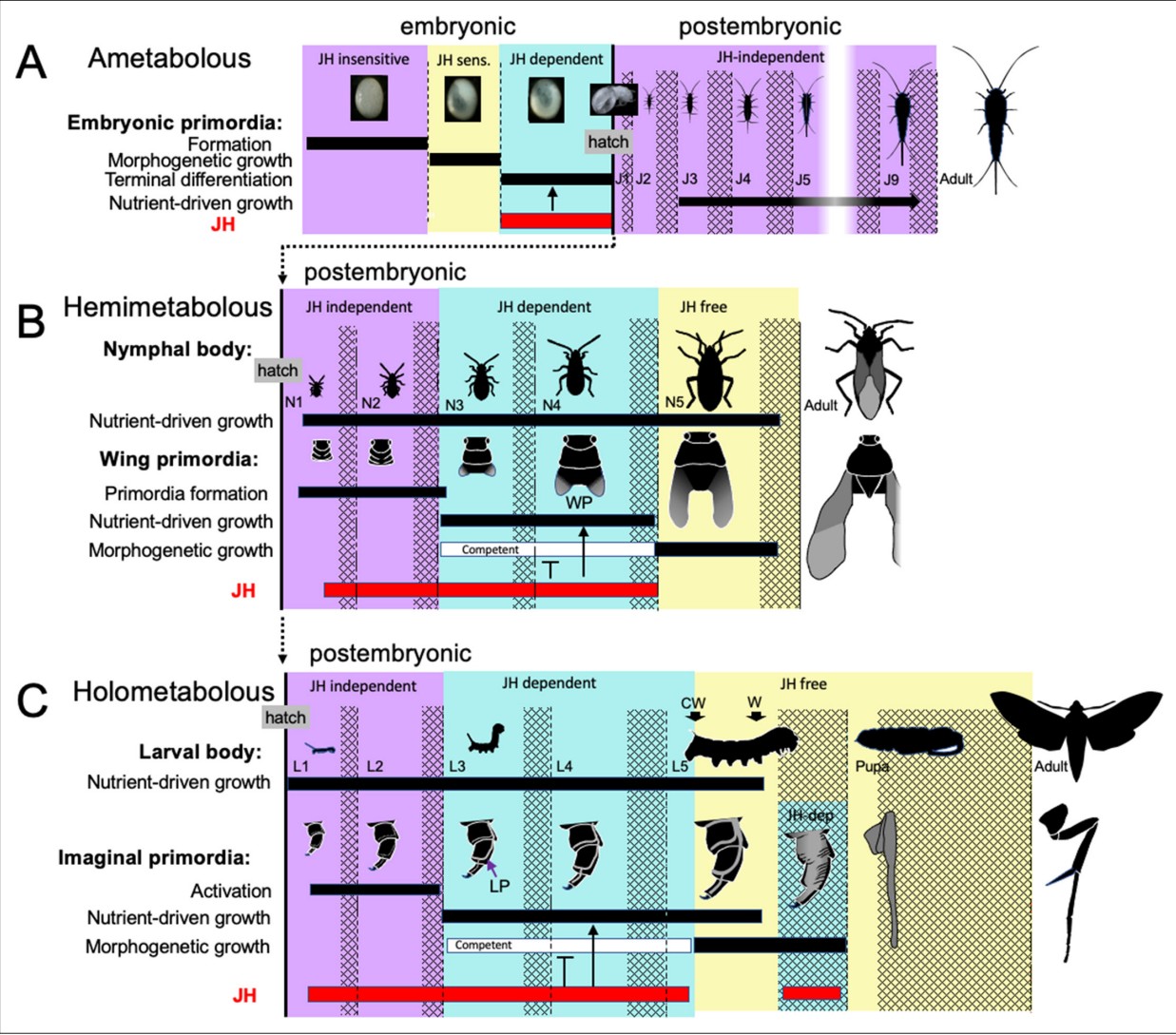

**Figure 12.** Comparison of the developmental effects of JH in *Thermobia* with its postembryonic actions in hemimetabolous and holometabolous insects. (**A**) As summarized in *Figure 10*, early *Thermobia* embryogenesis is JH insensitive, but JH sensitivity is acquired just prior to katatrepsis and the production of the first embryonic cuticle. JH appears after definitive dorsal closure and is required for the terminal differentiation of the juvenile. Once the juvenile form is established, JH is not then needed for its maintenance. (**B**) The early nymphal instars of hemimetabolous insects are also JH independent. Morphogenesis, associated with wing development, begins around the third nymphal stage and is correlated with the appearance of JH dependence. JH allows wing pad (WP) growth but suppresses its morphogenesis until the disappearance of JH ushers in the JH-free period which allows wing morphogenesis and differentiation. (**C**). The function of JH in holometabolous larvae is like that seen in nymphs except that imaginal primordia (like the leg primordia [LP]) are widely spread through the larval body and involve almost all of the adult organs. The imaginal primordia and the larval cells differ in their JH requirements during the prepupal period (see text). The pupal-adult transition then requires the absence of JH. CW: critical weight checkpoint; W: wandering stage; double crosshatch: period of new cuticle production preceding each ecdysis.

JH-independent to a JH-dependent state around their 3d instar. After this point, JH is needed to suppress their entry into metamorphosis (e.g. *Smykal et al., 2014*; *Chafino et al., 2019*). Although likely associated with growth (*Smykal et al., 2014*), the factors that cause metamorphic competence are unknown.

Nymphs and larvae differ from juveniles of *Thermobia* in that they have a second round of morphogenesis during their postembryonic growth (*Figure 12B and C*). However, the primordia that undergo postembryonic morphogenesis are typically dormant at hatching and need a period of growth before they start developing. In true bugs, like *Pyrrhocoris,* for example, wing pads only appear at the start of the 3rd nymphal stage (*Smykal et al., 2014*). Likewise in holometabolous larvae, such as the butterfly, *Pieris rapae*, cells of the wing primordia are evident at hatching but are still attached to larval cuticle.

As the larva feeds and grows, the cells of the primordium increase in size, detach from the cuticle, form an invaginated sac, and finally start dividing during the molt to the third larval stage (*Mercer, 1900*; see review in *Svácha, 1992*). How growth may cause the activation of dormant primordia has been examined for the eye-antenna (*Kenyon et al., 2003*) and the wing-notum discs (*Rafel and Milán, 2008*) of *Drosophila*. These discs are dormant at hatching because of opposing actions of short-range morphogens at the two edges of their respective discs. The inhibition imposed by these morphogens prevents the subsequent regionalization of these discs into eye *versus* antenna and wing *versus* notum domains, respectively. As the hatchling feeds and grows, the increase in disc cell size eventually moves the edges of the discs out of the zone of inhibition, thereby allowing the start of cell division and patterning (*Kenyon et al., 2003*; *Rafel and Milán, 2008*). Similar growth dependent changes in local morphogen signaling likely also control the initiation of wing pad formation in hemimetabolous nymphs. Whether the developing wing pads and other imaginal primordia are sources of the competence signals, though, is not known. However, once these tissues acquire metamorphic competence, JH is needed to delay its realization until nymphal/larval growth is over (*Smykal et al., 2014*).

The appearance of high levels of JH during postembryonic life likely occurred after the invention of wings and powered flight (*Figure 10B*). Fossil series of juvenile archaic flying insects, such as the Megasecoptera, show that the thorax of young juveniles bore small, articulated wings, although the latter were too small for flight (*Kukalova-Peck, 1978*; *Haug et al., 2016*). Through successive instars, the small wings showed positive allometric growth until they became large enough to support flight in the late juvenile and the adult stages. These insects, though, were still considered ametabolous because there was not an abrupt morphological change going from the juvenile to the adult (*Kukalova-Peck, 1978*). Wing development was subsequently modified so wing primordia appeared as immobile wing pads appressed to the body rather than as lateral, moveable winglets. As a wing pad, developing tissues were protected during juvenile growth and they deferred their morphogenesis and differentiation into wings until juvenile growth was finished. This wing pad is a feature of all extant hemimetabolous insects.

We assume that, like *Thermobia*, the ametabolous archaic winged insects produced little or no JH through juvenile life (*Figure 11B*). The change to making a wing pad, rather than a wing, may have occurred by bringing JH into the postembryonic realm at the time when wing morphogenesis was starting. While agents that drive morphogenesis and differentiation were separated in time during embryogenesis, their co-occurrence during wing formation may have forced a developmental compromise. The influence of local morphogens on the wing pad is evident in their cells remaining diploid, undergoing progressive patterning, and showing positive allometric growth, but under the influence of JH the wing pad cells remain undetermined, they make nymphal cuticle, and their growth depends on nutrient input. In modern insects, this compromise condition is tonically mainatined by JH. If JH or its effector Kr-h1 are removed, then the wing pad rapidly undergoes morphogenesis and differentiation into a wing (e.g. *Konopova et al., 2011*). Once wing pad formation established the need for JH in the postembryonic domain, JH was likely adopted by other systems, such as those regulating cuticle type, to control other phenotypic differences between the nymph and the adult.

Holometabolous larvae similarly use JH to suppress wing morphogenesis during larval growth (*Figure 12C*), but, in addition, they have scattered imaginal primordia that are embedded in larval organs or are early invaginated discs. The morphogenesis of these imaginal primordia is also suppressed by JH. When JH finally disappears at the critical weight checkpoint (*Nijhout and Williams, 1974*; *Nijhout, 1994*), the primordia embark on their program of morphogenetic growth that is independent of nutrient input (*Truman et al., 2006*).

The transformation of the larva into the adult requires two molts, with the nonfeeding pupal stage interposed in between. In terms of their JH requirements, the transition from the pupa to the adult is like that from the last stage nymph to the adult. They both occur in a JH-free environment, and exposure to JH at the outset of the molt redirects development to make another pupal or nymphal stage, respectively. The transition from the larva to the pupa, though, is unusual in that JH returns during the prepupal period (*Figure 11A*). As depicted in *Figure 12C*, this JH is especially important for the tissues that come from the imaginal primordia. For regular larval tissues, the commitment from the larval program to the pupal program is caused by the small ecdysteroid peak that evokes wandering and, thereafter, JH cannot stop their pupal differentiation (*Riddiford, 1976*). The imaginal primordia,

by contrast, have an earlier commitment to pupal differentiation at the critical weight checkpoint. In a species- and tissue-specific fashion, these imaginal primordia may fail to undergo normal pupal differentiation if JH is absent during the prepupal period (*Williams, 1961*; *Riddiford et al., 2010*). This requirement of some imaginal primordia for JH for their proper pupal differentiation is reminiscent of the requirement by embryonic primordia of basal insects for JH to complete their nymphal differentiation. It is important, though, that the outcome of the lack of JH differs in the two cases: its lack in the embryo results in suppression of juvenile/nymphal differentiation, whereas its lack in the prepupa results in an 'overshoot' to the adult stage. We do not know enough about either of these phenomena, though, to reconcile these differences.

The above discussion proposes an inherent antagonism between JH and morphogen signaling. This antagonism has been assumed based on phenotypic effects such as JH stopping eye primordium patterning in embryos (*Figure 6—figure supplement 1*) or in premetamorphic larvae (*Nijhout and Kremen, 1998*). To date, though, the only morphogen that has been directly tied to JH is the circulating morphogen, myoglianin. A major source of myoglianin in larvae is muscle and it provides a proxy for body size, to inform the insects that sufficient growth has occurred to permit metamorphosis (*He et al., 2020*). Studies on the cricket, *Gryllus bimaculatus* (*Ishimaru et al., 2016*) and on *Blattella* (*Kamsoi and Belles, 2019*) show that myoglianin acts on the corpora allata to suppress JH production, thereby allowing imaginal primordia to switch to morphogenetic growth. Late in larval life in *Drosophila*, systemic myoglianin also provides information to coordinate imaginal disc growth to overall body size (*Upadhyay et al., 2020*). In *Thermobia* embryos, we find that levels of JH and myoglianin are also related, but the relationship is reversed, with myoglianin appearing first, during the period of morphogenetic growth, and early application of JH able to suppress this appearance. We have no experimental data on the function of myoglianin in *Thermobia* embryos, but in a short germband insect where yolk content might be variable, it may be important to have a circulating morphogen to match appendage size to body size. Although the inhibitory relationships between myoglianin and JH are in opposite directions in the embryo versus in the juvenile, an interaction of the two signaling systems appears to extend deep into insect evolution.

## Conclusions

The ancestral developmental role for JH appears to have been in the embryo where it functioned to promote terminal tissue differentiation. JH also had powerful morphostatic activity, but this activity was not evident during normal development because JH appeared late in embryogenesis after morphogenesis had essentially finished. These embryonic actions of JH, though, preadapted it to become the consummate *status quo* hormone in the postembryonic realm. Once *Thermobia* embryos have hatched, they do not need JH to maintain their juvenile condition. The same is true of nymphs and larvae until around the time that dormant primordia initiate their morphogenetic growth. Once these primordia acquire competent to begin programs of morphogenesis, the morphostatic actions of JH maintain them in an immature condition. The removal of JH is needed for these primordia to achieve their developmental potential.

JH signaling and action extend back into the Crustacea where its immediate precursor, methyl farnesoate (MF), is involved in reproductive control (*Borst et al., 1987*). MF also affects crustacean development, but these effects are subtle and diverse (*Laufer and Biggers, 2001*) and difficult to compare with JH effects in *Thermobia* at this time. An involvement of farnesol derivatives in growth and differentiation may extend beyond the pancrustacea. In the yeast *Candida albicans*, secreted farnesoic acid acts as a 'quorum sensing signal' that controls the transition from a budding form to a hyphal form (*Hornby et al., 2001*; *Oh et al., 2001*). Amongst animals, the genes for the enzymes to synthesize farnesoic acid are widely found throughout invertebrate phyla (*So et al., 2022*). In mammals, farnesol stimulates differentiation in epidermal keratinocytes (*Hanley et al., 2000*) and it also suppresses xenograft-based tumor growth in mice (*Lee et al., 2015*). Indeed, farnesol-based molecules may have an ancient involvement in switching cells between developmental states and this capacity may have been exploited by the insects to provide the hormonal system that regulates their metamorphosis.

## Methods

Information on specialized reagents is summarized in *Table 1*.

**Table 1.** Specialized animals and reagents.

| Reagent / animal | Source | Catalog number |
| --- | --- | --- |
| Thermobia domestica | 30+year laboratory colony | est. from wild caught |
| pyriproxyfen | Sumitomo Co | #95737-68-1 |
| 7 ethoxyprecocene | Sigma Aldrich | #65383-73-5 |
| DAPI | Sigma Aldrich | #28718-90-3 |
| Calcofluor White | Sigma Aldrich | #4193-55-9 |
| Propidium Iodide | Thermo Fisher Scientific | #440301000 |
| Oregon Green phalloidin | Thermo Fisher Scientific | #O7466 |
| rabbit anti-distal-less | gift from Grace Panganiban | |
| rabbit anti P-HistoneH3 (ser 10) | EDM Millipore | #06–570 |
| Normal Donkey Serum | Jackson ImmunoResearch | #017-000-121 |
| AF488 Donkey anti-rabbit IgG | Jackson ImmunoResearch | #711-545-152 |
| AF594 Donkey anti-rabbit IgG | Jackson ImmunoResearch | #711-585-152 |
| DPX mountant | Electron Microscopy Sciences | # 13512 |

## Animals

The firebrats, *Thermobia domestica*, were reared at 37 °C in small, covered polyethylene boxes. Layers of corrugated filter paper provided surfaces for the insects to move on and hide. The insects were fed dry Gerber's baby rice cereal and a few pellets of dry cat food. Water was available in cotton-plugged vials. The incubator was kept above 70% relative humidity with a saturated KCl solution.

Cotton balls were placed between the filter paper sheets for oviposition. For timed egg collections, the cotton balls were replaced morning and evening to achieve 12 hr egg collections, and each was stored in a covered petri dish at 37 °C until needed. The age of the embryos was referenced to the midpoint of the collection period.

## Juvenile hormone titers

Timed egg collections were extracted based on the method of *Bergot et al., 1981b* as modified by Lacy Barton and Justina Sanny (personal communication) and by LMR. Briefly, 200 eggs were homogenized in 350 µl acetonitrile containing 31.25 pg deuterated JH III (JHD3) in a silanized 2 ml Snap-It vial (Thermo-Fisher) on ice using a Benchmark D1000 homogenizer (Benchmark Scientific, New York), then capped and spun at 5000 × *g* for 15 min at 4 °C. The supernatant was transferred to a new silanized vial. The pellet was resuspended in 250 µl acetonitrile containing 31.25 pg JHD3 III, spun at 5000 × *g* for 15 min, then the two supernatants combined. The combined supernatants were extracted with two volumes n-pentane and 4 volumes of 4% NaCl, then spun at 2000 × *g* for 20 min at 4 °C. The organic phases and aqueous phase were put into separate silanized vials, and the aqueous phase re-extracted with a half volume of n-pentane and spun 20 min at 4 °C. The resultant organic phase was combined with the original organic phases which was then washed with an equal volume of 4% NaCl, and centrifuged at 2000 × *g* for 20 min at 4 °C. The organic phase was then transferred to a new silanized vial, the solvent gently blown off under a stream of nitrogen, and the residue resuspended in 200 µl acetonitrile. The residue and two 150 µl rinses of the vial were loaded onto a Sep-Pak C18 column (3 cc; Waters Corp., Milford, MA, USA). The purified extract was eluted from the column with 2 ml acetonitrile and frozen at –20 °C until analysis by the liquid chromatography-tandem mass spectrometry method of *Ramirez et al., 2020*.

## Ecdysteroid titers

Ecdysteroid concentrations in timed egg collections were measured using the 20-Hydroxyecdysone (20E) Enzyme Immunoassay (EIA) kit (Cayman Chemical Company, Ann Arbor, MI) based on the method developed by *Porcheron et al., 1989*. Staged *Thermobia* embryos (200 per sample) were

**Table 2.** Primers used for real-time PCR.

| gene | sequence accession number (NCBI) | forward primer (5'–3') | reverse primer (5'–3') |
|------|----------------------------------|------------------------|------------------------|
| Met | JN416986.1 | TACTCCATCCACACAGTCAAGG | TTCCGTGATTCGACGATCTCTC |
| Kr-h1 | JN416989.1 | ACTCCGTCGAATGGTACTAGTG | GTTCTTGCATGGGAGGAAACTG |
| myo | GASN02042720.1 | TTCAACAAGCAAGCCCACAAG | ACACCGATCCACTACCATTCC |
| rp49 | AB689035.1 | CTAAAGAGGAACTGGCGCAAAC | GTTTAGTCTTCTTGGCGCTTCC |

homogenized in 250 µls ice cold 75% aqueous methanol as outlined in *Margam et al., 2006*. Homogenized samples were centrifuged for 15 min at 13,000 × *g* and 4 °C. Supernatants were transferred to 1.5 ml microcentrifuge tubes on ice. Pellets were resuspended in 250 µls 75% aqueous methanol, vortexed, placed on ice for 30 min, then centrifuged as before, and the supernatants pooled. One quarter of the volume was dried down under nitrogen, stored desiccated at –20 °C. It was resuspended in EIA buffer and diluted 1/100 just before assay. Following the kit instructions, the EIA was performed in a 96-well microtiter plate and is based on the competition between 20E (in standard or sample) and acetylcholinesterase-labeled 20E for the ecdysteroid antiserum that has been bound to the IGG-coated well plates. The plate was incubated with gyration overnight at 4 °C, then washed five times with EIA buffer. The enzymatic substrate for acetylcholinesterase and chromogen, which form a yellow compound that absorbs between 405–414 nm, was then added to each well and incubated at room temperature with gyration for 60 min. The intensity of the color is inversely proportional to the amount of ecdysteroid present as determined by spectrophotometry on a µQuant Microplate Spectrophotometer plate reader at 410 nm (BioTek Instruments, Vermont). The kit solutions were resuspended in UltraPure water (Cayman Chemical). The readings were compared to a standard curve derived from serial dilutions of the 20E standard and quality control provided by the kit.

## Bioactive compounds
Pyriproxyfen (Sumitomo Co.) and 7-ethoxyprecocene (Sigma Aldrich) were dissolved in cyclohexane (HPLC grade, Sigma Aldrich) and stored at –20 °C. Using a 10 µl Hamilton syringe, 0.2 µl of hormone solution or of cyclohexane was applied with a repeating dispenser to staged eggs on double-stick tape.

## RNA analysis
Collections of timed embryos were placed in a 0.5 ml Eppendorf tube to which 150 µl Trizol (Ambion) was added. The tube was vortexed, briefly spun, then frozen in liquid nitrogen. Total RNA was isolated using TRIzol (Invitrogen), treated with TURBO DNase (Invitrogen) and purified with RNA Clean & Concentrator (Zymo Research). cDNA was prepared from 2 µg of the RNA using oligo-dT primer and SuperScript IV First-Strand Synthesis System (Invitrogen). PCR was performed using SsoAdvanced Universal SYBR Green Supermix (Bio-Rad); each 20 µl reaction contained 100 ng cDNA and forward and reverse primers, each primer at a final concentration of 500 nM. Reactions were run in the CFX Connect Real-Time PCR Detection System (Bio-Rad) and normalized expression was calculated using the associated software according to comparative Ct method (*Schmittgen and Livak, 2008*). The expression levels of each target gene were normalized against expression levels of the ribosomal protein 49 (rp49). Primers were designed using Primer3Plus (*Untergasser et al., 2012*) on the gene sequences that were derived from NCBI: Met and Kr-h1 were isolated by BK previously (*Konopova et al., 2011*), myo was found in the TSA database by blast search using *Drosophila* myo as a query and the amplicon sequence was also checked in the Thermobia genome (*Brand et al., 2018*) to check for nucleotide mismatches (sites of polymorphism). The accession numbers and primer sequences are in *Table 2*. Melting curves were examined after each run and for each pair of primers several finished runs were visualized on a 2% agarose gel; only a single product was detected.

On selected samples including young embryos, old embryos, and juveniles we checked which of the possible reference genes would give the most consistent expression levels: we tested rp49 (also known as Ribosome protein L32), Elongation factor-1α, Actin 5 C (β-actin), α– Tub84B (α-tubulin).

rp49 appeared most suitable. This choice agrees with other studies in *Thermobia* (*Bai et al., 2020*; *Fernandez-Nicolas et al., 2023*).

## Immunocytochemistry

Embryos were dissected and fixed in 3.9% formaldehyde (Thermo Fisher Scientific) in phosphate-buffered saline (PBS; Fisher Scientific) for 30–60 min, then rinsed three times in PBS with1% Triton X100 (PBS-TX; Thermo Fisher Scientific). They were then incubated with agitation in PBS-TX with various combinations of stains for 1–2 days at 4 °C. Tissue stains were propidium iodide (Thermo Fisher Scientific, 1:1000 dilution of 1 mg/ml stock) or DAPI (Thermo Fisher Scientific, 1:1000 dilution of 1 mg/ml stock) for DNA, Alexa-488 conjugated phalloidin (Thermo Fisher Scientific, 1:100 dilution of 200 units/ml stock in methanol) for actin, and Calcfluor White (Sigma Chemical) for cuticle. Stained tissues were repeatedly rinsed, mounted on poly-L-lysine (Sigma-Aldrich) coated coverslips, dehydrated through a graded ethanol series, cleared through three changes of xylene, and mounted in DPX (Sigma-Aldrich).

A rabbit antibody against phosphohistone H3 (#06–570: anti phosphor-Histone H3 (ser 10); EDM Millipore, Darmstadt, Germany) was used at a 1:1000 dilution to identify mitotic cells. Tissue was preblocked in 2% normal donkey serum (Jackson Immunoresearch Laboratories, West Grove, PA, USA) for 15–30 min, then incubated with the primary antibody over two to three nights. Following five to six rinses in PBS-TX, tissues were incubated with a secondary antibody along with a variety of stains. Secondary antibodies were various Alexa Fluor 488-, 594-, or 647-conjugated donkey antisera raised against rabbit IgG fractions and used at 1:500 (Jackson Immunoresearch Laboratories, West Grove, PA, USA). Selection of the secondary depended on the combination of stains that were used. After staining, tissues were mounted, dehydrated, cleared and mounted as described above. The preparations were imaged on a Zeiss 800 confocal microscope and processed with ImageJ software (http://imagej.nih.gov/ij/).

## Acknowledgements

We thank Helen Chea who generated the early pilot data on JH effects on *Thermobia* embryos during a Research Apprenticeship Course at Friday Harbor Laboratories in 2003. The unpublished studies on *Schistocerca* were carried out while JWT and LMR were on sabbatical leave in the lab of Prof. Michael Akam at the University of Cambridge. The research was supported by a NIH-NIAID R21 award (R21AI167849) to FGN, and project grant 22–21244 S from the Czech Science Foundation, Czech Republic to MN. JWT and LMR were supported by a gift to UW from the Howard Hughes Medical Institute.

## Additional information

### Funding

| Funder | Grant reference number | Author |
| --- | --- | --- |
| Howard Hughes Medical Institute | Gift to UW | James W Truman Lynn M Riddiford |
| National Institutes of Health | R21 AI167849 | Fernando G Noriega |
| Czech Science Foundation | 22-21244S | Marcela Nouzova |

The funders had no role in study design, data collection and interpretation, or the decision to submit the work for publication.

### Author contributions

James W Truman, Conceptualization, Formal analysis, Investigation, Methodology, Writing – original draft, Writing – review and editing; Lynn M Riddiford, Conceptualization, Methodology, Writing – review and editing; Barbora Konopova, Marcela Nouzova, Fernando G Noriega, Formal analysis,

Investigation, Methodology, Writing – review and editing; Michelle Herko, Formal analysis, Investigation, Methodology

## Author ORCIDs
James W Truman https://orcid.org/0000-0002-9209-5435
Lynn M Riddiford https://orcid.org/0000-0002-3026-7430

Reviewer #1 (Public Review): https://doi.org/10.7554/eLife.92643.3.sa1
Reviewer #2 (Public Review): https://doi.org/10.7554/eLife.92643.3.sa2
Reviewer #3 (Public Review): https://doi.org/10.7554/eLife.92643.3.sa3
Author response https://doi.org/10.7554/eLife.92643.3.sa4

## Additional files

### Supplementary files
• MDAR checklist

### Data availability
All data generated or analysed during this study are included in the manuscript and supporting files. Source data are provided for Figures 1, 2, 4, 6, and 7. Figure 1—source data 1 provides data used to generate the figure. Figure 2—source data 1 gives data used to juvenile hormone titer during embryogenesis (Figure 2A). Figure 2—source data 2 gives assay data used to generate ecdysteroid levels during embryogenesis. Figure 4—source data 1 provides assay data that determined the effectiveness of precocene amd JHm treatment during embryogenesis. Figure 6—source data 1 provides data showing the subsequent development after treatment with JHm at various times of embryogenesis. Figure 7—source data 1 provides the quantiattive data of pHH3 labeled cells in developing limbs after treatment with JHm at different developmental times.

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
