## [Editor Report · eLife assessment]

This **important** study presents findings regarding the role of Juvenile Hormone in development and cell differentiation in the ametabolous insect Thermobia domestica, providing an in-depth analysis of JH's roles in a member of this basally branching group. The evidence supporting the claims of the authors is **convincing**, drawing on a broad range of approaches and variety of experimental techniques. While the interpretation of this work in the wider context - its relevance for the evolution of metamorphosis - is in some places somewhat speculative, the work will be of interest to evolutionary developmental biologists studying the evolution of metamorphosis, and the evolution of insects in general.

---

## [Referee Report · Reviewer #1 (Public Review)]

Summary:

This paper provides strong evidence for the roles of JH in an ametabolous insect species. In particular, it demonstrates that:

• JH shifts embryogenesis from a growth mode to a differentiation mode and is responsible for terminal differentiation during embryogenesis. This, and other JH roles, are first suggested as correlations, based on the timing of JH peaks, but then experimentally demonstrated using JH antagonists and rescue thereof with JH mimic. This is a robust approach and the experimental results are very convincing.

• JH redirects ecdysone-induced molting to direct formation of a more mature cuticle

• Kr-h1 is downstream of JH in Thermobia, as it is in other insects, and is a likely mediator of many JH effects

• The results support the proposed model that an ancestral role of JH in promoting and maintaining differentiation was coopted during insect radiations to drive the evolution of metamorphosis. However, alternate evolutionary scenarios should also be considered.

Strengths:

Overall, this is a beautiful, in-depth student. The paper is well-written and clear. The background places the work in a broad context and shows its importance in understanding fundamental questions about insect biology. The researchers are leaders in the field, and a strength of this manuscript is their use of a variety of different approaches (enzymatic assays, gene expression, agonists & antagonists, analysis of morphology using different types of microscopy and detection, and more) to attack their research questions. The experimental data is clearly presented and carefully executed with appropriate controls and attention to detail. The 'multi-pronged' approach provides support for the conclusions from different angles, strengthening conclusions. In sum, the data presented are convincing and the conclusions about experimental outcomes are well-justified based on the results obtained.

Weaknesses:

This paper provides more detail than is likely needed for readers outside the field but also provides sufficient depth for those in the field. This is both a strength and a weakness. I would suggest the authors shorten some aspects of their text to make it more accessible to a broader audience. In particular, the discussion is very long and accompanied by two model figures. The discussion could be tightened up and much of the text used for a separate review article (perhaps along with Figure 11) that would bring more attention to the proposed evolution of JH roles.

---

## [Referee Report · Reviewer #2 (Public Review)]

The authors have studied in detail the embryogenesis of the ametabolan insect Thermobia domestica. They have also measured the levels of the two most important hormones in insect development: juvenile hormone (JH) and ecdysteroids. The work then focuses on JH, whose occurrence concentrates in the final part (between 70 and 100%) of embryo development. Then, the authors used a precocene compound (7-ethoxyprecocene, or 7EP) to destroy the JH producing tissues in the embryo of the firebrat T. domestica, which allowed to unveil that this hormone is critically involved in the last steps of embryogenesis. The 7EP-treated embryos failed to resorb the extraembryonic fluid and did not hatch. More detailed observations showed that processes like the maturational growth of the eye, the lengthening of the foregut and posterior displacement of the midgut, and the detachment of the E2 cuticle, were impaired after the 7EP treatment. Importantly, a treatment with a JH mimic subsequent to the 7EP treatment restored the correct maturation of both the eye and the gut. It is worth noting that the timing of JH mimic application was essential for correcting the defects triggered by the treatment with 7EP.

This is a relevant result in itself since the role of JH in insect embryogenesis is a controversial topic. It seems to have an important role in hemimetabolan embryogenesis, but not so much in holometabolans. Intriguingly, it appears important for hatching, an observation made in hemimetabolan and in holometabolan embryos. Knowing that this role was already present in ametabolans is relevant from an evolutionary point of view, and knowing exactly why embryos do not hatch in the absence of JH, is relevant from the point of view of developmental biology.

Then, the authors describe a series of experiments applying the JH mimic in early embryogenesis, before the natural peak of JH occurs, and its effects on embryo development. Observations were made under different doses of JHm, and under different temporal windows of treatment. Higher doses triggered more severe effects, as expected, and different windows of application produced different effects. The most used combination was 1 ng JHm applied 1.5 days AEL, checking the effects 3 days later. Of note, 1.5 days AEL is about 15% embryonic development, whereas the natural peak of JH occurs around 85% embryonic development. In general, the ectopic application of JHm triggered a diversity of effects, generally leading to an arrest of development. Intriguingly, however, a number of embryos treated with 1 ng of JHm at 1.5 days AEL showed a precocious formation of myofibrils in the longitudinal muscles. Also, a number of embryos treated in the same way showed enhanced chitin deposition in the E1 procuticle and showed an advancement of at least a day in the deposition of the E2 cuticle.

While the experiments and observations are done with great care and are very exhaustive, I am not sure that the results reveal genuine JH functions. The effects triggered by a significant pulse of ectopic JHm when the embryo is 15% of the development will depend on the context: the transcriptome existing at that time, especially the cocktail of transcription factors. This explains why different application times produce different effects. This also explains why the timing of JHm application was essential for correcting the effects of 7EP treatment. In this reasoning, we must consider that the context at 85% development, when the JH peaks in natural conditions and plays its genuine functions, must be very different from the context at 15% development, when the JHm was applied in most of the experiments. In summary, I believe that the observations after the application of JHm reveal effects of the ectopic JHm, but not necessarily functions of the JH. If so, then the subsequent inferences made from the premise that these ectopic treatments with JHm revealed JH functions are uncertain and should be interpreted with caution.

Those inferences affect not only the "JH and the progressive nature of embryonic molts" section, but also, the "Modifications in JH function during the evolution of hemimetabolous and holometabolous life histories" section, and the entire "Discussion". In addition to inferences built on uncertain functions, the sections mentioned, especially the Discussion, I think suffer from too many poorly justified speculations. I love speculation in science, it is necessary and fruitful. But it must be practiced within limits of reasonableness, especially when expressed in a formal journal.

Finally, In the section "Modifications in JH function during the evolution of hemimetabolous and holometabolous life", it is not clear the bridge that connects the observations on the embryo of Thermobia and the evolution of modified life cycles, hemimetabolan and holometabolan.

---

## [Referee Report · Reviewer #3 (Public Review)]

Summary:

In this manuscript, the authors use inhibitors and mimetics of juvenile hormone (JH) to demonstrate that JH has a key role in late embryonic development in Thermobia, specifically in gut and eye development but also resorption of the extraembryonic fluid and hatching. They then exogenously apply JH early in development (when it is not normally present) to examine the biological effects of JH at these stages. This causes a plethora of defects including developmental arrest, deposition of chitin, limb development, and enhanced muscle differentiation. The authors interpret these early effects on development as JH being important for the shift from morphogenetic growth to differentiation - a role that they speculate may have facilitated the evolution of metamorphosis (hemi- and holo-metaboly). This paper will be of interest to insect evo-devo researchers, particularly those with interests in the evolution of metamorphosis.

Strengths:

The experiments are generally conducted very well with appropriate controls and the authors have included a very detailed analysis of the phenotypes.

The manuscript significantly advances our understanding of Thermobia development and the role of JH in Thermobia development.

The authors interpret this data to present some hypotheses regarding the role of JH in the evolution of metamorphosis, some aspects of which can be addressed by future studies.

Weaknesses:

The results are based on using inhibitors and mimetics of JH and there was no attempt to discern immediate effects of JH from downstream effects. The authors show, for instance, that the transcription of myoglianin is responsive to JH levels, it would have been interesting to see if any of the phenotypic effects are due to myoglianin upregulation/suppression (using RNAi for example). These kinds of experiments will be necessary to fully work out if and how the JH regulatory network has been co-opted into metamorphosis.

The results generally support the authors' conclusions. However, the discussion contains a lot of speculation and some far-reaching conclusions are made about the role of JH and how it became co-opted into controlling metamorphosis. There are some interesting hypotheses presented and the author's speculations are consistent with the data presented. However, it is difficult to make evolutionary inferences from a single data point as although Thermobia is a basally branching insect, the lineage giving rise to Thermobia diverged from the lineages giving rise to the holo- and hemimetabolous insects approx.. 400 mya and it is possible that the effects of JH seen in Thermobia reflect lineage-specific effects rather than the 'ancestral state'. The authors ignore the possibility that there has been substantial rewiring of the networks that are JH responsive across these 400 my. I would encourage the authors to temper some of the discussion of these hypotheses and include some of the limitations of their inferences regarding the role of JH in the evolution of metamorphosis in their discussion.

---

## [Author Response]

The following is the authors’ response to the original reviews.

We appreciate the care and the detail shown by the Reviewers. Their comments have made our article more focused and more accessible to a general audience.

We would like to begin with a comment about the last sentence of the “eLife assessment”. The evolution of metamorphosis in insects was a major triumph in animal evolution that subsequently impacted almost every aspect of plant and animal evolution in the terrestrial and freshwater aquatic biospheres. Unlike the metamorphoses of most other groups, whose evolutions are lost in time, insect evolution arose relatively recently (~400 mya) and insect orders have branched off at various points in this evolution and have persisted to modern times. Although these “relic” groups also have undergone millions of years of evolution and specialization, they still provide us with windows into how this progression may have come about. The study of these groups provides a unique opportunity to explore the mechanisms that underlie major life history shifts and should be of interest to anyone interested in evolution – not just entomologists.

**Reviewer #1 (Public Review):**
Summary:This paper provides strong evidence for the roles of JH in an ametabolous insect species. In particular, it demonstrates that:• JH shifts embryogenesis from a growth mode to a differentiation mode and is responsible for terminal differentiation during embryogenesis. This, and other JH roles, are first suggested as correlations, based on the timing of JH peaks, but then experimentally demonstrated using JH antagonists and rescue thereof with JH mimic. This is a robust approach and the experimental results are very convincing.• JH redirects ecdysone-induced molting to direct formation of a more mature cuticle• Kr-h1 is downstream of JH in Thermobia, as it is in other insects, and is a likely mediator of many JH effects• The results support the proposed model that an ancestral role of JH in promoting and maintaining differentiation was coopted during insect radiations to drive the evolution of metamorphosis. However, alternate evolutionary scenarios should also be considered.Strengths:Overall, this is a beautiful, in-depth student. The paper is well-written and clear. The background places the work in a broad context and shows its importance in understanding fundamental questions about insect biology. The researchers are leaders in the field, and a strength of this manuscript is their use of a variety of different approaches (enzymatic assays, gene expression, agonists & antagonists, analysis of morphology using different types of microscopy and detection, and more) to attack their research questions. The experimental data is clearly presented and carefully executed with appropriate controls and attention to detail. The 'multi-pronged' approach provides support for the conclusions from different angles, strengthening conclusions. In sum, the data presented are convincing and the conclusions about experimental outcomes are well-justified based on the results obtained.Weaknesses:This paper provides more detail than is likely needed for readers outside the field but also provides sufficient depth for those in the field. This is both a strength and a weakness. I would suggest the authors shorten some aspects of their text to make it more accessible to a broader audience. In particular, the discussion is very long and accompanied by two model figures. The discussion could be tightened up and much of the text used for a separate review article (perhaps along with Figure 11) that would bring more attention to the proposed evolution of JH roles.

We appreciate the comments about the strengths and weaknesses of the paper. To deal with the weaknesses, we have condensed some of the Results to make them less cumbersome and the Discussion has been completely revised, keeping a sharp focus on the actions of JH in Thermobia embryos and how these actions relate to the status quo functions of JH in insects with metamorphosis. As part of the revision of the Discussion, we have replaced Figures 10 and 11.

**Reviewer #1 (Recommendations For The Authors):**
In keeping with my public review, this paper is very strong and I have very few suggestions for improvement. They are:(1) Thermobia are extant insects and are not ancestral insects. It is likely that they retain features found in an insect ancestor. However, these insects have been evolving for a very long time, and for any one feature, many changes may have occurred, both gain and loss of gene function and morphology. Further, even for morphological features present in an extant species that are the same as an ancestor, genetic pathways regulating this feature may have changed over time (see for examples papers from the Haag and Pick labs). Although I realize this is a small, possibly almost semantic point, I feel it is important to be precise here. For example, in the title, "before" is speculative as there could have been a different role in the ancestor with the role in embryogenesis arising in lineages leading to Thermobia; similarly in the abstract, "this ancestral role of JH' is an overstatement since we cannot actually measure the ancestral role.

Since the title has already been cited in a Perspectives review, we decided to keep the title as is.

(2) I don't understand the results in Met and myo in Fig. 3B. Perhaps include them in the explanation of Fig.3 and not after the description of Fig. 4 and explain them in more detail (or perhaps not include them at all?). I don't really understand the statistical analysis of these panels either.

We have revised the figure legends to explain the statistics.

(3) Another point regarding language - talking about the embryo being "able" to go through a developmental stage implies decision-making. I would suggest dropping that wording (e.g, in the description of Fig. 5C). Similarly, in explaining Fig. 6B, it would be more correct to say "JH treatment no longer inhibited" than as written "could no longer inhibit" (implying 'no matter how hard it tried, it still couldn't do it')

We have removed the “can’t” wording. Figure 6 has been revised

**Reviewer #2 (Public Review):**
The authors have studied in detail the embryogenesis of the ametabolan insect Thermobia domestica. They have also measured the levels of the two most important hormones in insect development: juvenile hormone (JH) and ecdysteroids. The work then focuses on JH, whose occurrence concentrates in the final part (between 70 and 100%) of embryo development. Then, the authors used a precocene compound (7-ethoxyprecocene, or 7EP) to destroy the JH producing tissues in the embryo of the firebrat T. domestica, which allowed to unveil that this hormone is critically involved in the last steps of embryogenesis. The 7EP-treated embryos failed to resorb the extraembryonic fluid and did not hatch. More detailed observations showed that processes like the maturational growth of the eye, the lengthening of the foregut and posterior displacement of the midgut, and the detachment of the E2 cuticle, were impaired after the 7EP treatment. Importantly, a treatment with a JH mimic subsequent to the 7EP treatment restored the correct maturation of both the eye and the gut. It is worth noting that the timing of JH mimic application was essential for correcting the defects triggered by the treatment with 7EP.This is a relevant result in itself since the role of JH in insect embryogenesis is a controversial topic. It seems to have an important role in hemimetabolan embryogenesis, but not so much in holometabolans. Intriguingly, it appears important for hatching, an observation made in hemimetabolan and in holometabolan embryos. Knowing that this role was already present in ametabolans is relevant from an evolutionary point of view, and knowing exactly why embryos do not hatch in the absence of JH, is relevant from the point of view of developmental biology.

The unique and intriguing aspect of juvenile hormone is its status quo action in the control of metamorphosis. Our reason for dealing with an insect group that branched off from the line of insects that eventually evolved metamorphosis, was to gain insight into the ancestral functions of this hormone. Our data from Thermobia as well as that from grasshoppers and crickets indicate that the developmental actions of JH were originally confined to embryogenesis where it promoted the terminal differentiation of the embryo. Its actions in promoting differentiation also included suppressing morphogenesis. This latter function was not pronounced during embryogenesis because JH only appeared after morphogenesis was essentially completed. However, it was a preadaptation that proved useful in more derived insects that delayed aspects of morphogenesis into the postembryonic realm. JH was then used postembryonically to inhibit morphogenesis until late in juvenile growth when JH disappears, and this inhibition is released.

Then, the authors describe a series of experiments applying the JH mimic in early embryogenesis, before the natural peak of JH occurs, and its effects on embryo development. Observations were made under different doses of JHm, and under different temporal windows of treatment. Higher doses triggered more severe effects, as expected, and different windows of application produced different effects. The most used combination was 1 ng JHm applied 1.5 days AEL, checking the effects 3 days later. Of note, 1.5 days AEL is about 15% embryonic development, whereas the natural peak of JH occurs around 85% embryonic development. In general, the ectopic application of JHm triggered a diversity of effects, generally leading to an arrest of development. Intriguingly, however, a number of embryos treated with 1 ng of JHm at 1.5 days AEL showed a precocious formation of myofibrils in the longitudinal muscles. Also, a number of embryos treated in the same way showed enhanced chitin deposition in the E1 procuticle and showed an advancement of at least a day in the deposition of the E2 cuticle.While the experiments and observations are done with great care and are very exhaustive, I am not sure that the results reveal genuine JH functions. The effects triggered by a significant pulse of ectopic JHm when the embryo is 15% of the development will depend on the context: the transcriptome existing at that time, especially the cocktail of transcription factors. This explains why different application times produce different effects. This also explains why the timing of JHm application was essential for correcting the effects of 7EP treatment. In this reasoning, we must consider that the context at 85% development, when the JH peaks in natural conditions and plays its genuine functions, must be very different from the context at 15% development, when the JHm was applied in most of the experiments. In summary, I believe that the observations after the application of JHm reveal effects of the ectopic JHm, but not necessarily functions of the JH. If so, then the subsequent inferences made from the premise that these ectopic treatments with JHm revealed JH functions are uncertain and should be interpreted with caution.

We disagree with the reviewer. An analogous situation would be in exploring gene function in which both gain-of-function and loss-of-function experiments often provide complementary insights into how a gene functions. We see JH effects only when its receptor, Met, is present and JH can induce its main effector protein, Kr-h1. The latter gives us confidence that we are looking at bona fide JH effects. We have also kept in mind, though, that the nature of the responding tissues is changing through time. Nevertheless, we see a consistent pattern of responses in the embryo and these can be related to its postembryonic effects in metamorphic insects.

Those inferences affect not only the "JH and the progressive nature of embryonic molts" section, but also, the "Modifications in JH function during the evolution of hemimetabolous and holometabolous life histories" section, and the entire "Discussion". In addition to inferences built on uncertain functions, the sections mentioned, especially the Discussion, I think suffer from too many poorly justified speculations. I love speculation in science, it is necessary and fruitful. But it must be practiced within limits of reasonableness, especially when expressed in a formal journal.

We have tried to dial back the speculation.

Finally, In the section "Modifications in JH function during the evolution of hemimetabolous and holometabolous life", it is not clear the bridge that connects the observations on the embryo of Thermobia and the evolution of modified life cycles, hemimetabolan and holometabolan.

Our Figure 12 should put this into perspective.

**Reviewer #2 (Recommendations For The Authors):**
Main points(1) Please, reduce the level of overinterpretation of ectopic treatment experiments with JHm, since the resulting observations represent effects, but not necessarily functions of JH.

We have revised this section to indicate that the “effects” of ectopic treatments provide insights into the function of JH. Using a genetic analogy, both “loss-of-function” and “gain-of-function” experiments provide insights into a given gene. (see response to Public Comments)

(2) Especially in the sections "JH and the progressive nature of embryonic molts" and "Modifications in JH function during the evolution of hemimetabolous and holometabolous life histories", and the entire "Discussion", please keep the level of speculation within reasonable limits, avoiding especially the inference of conclusions on the basis of speculation, itself based on previous speculation.

We have toned down some of the speculation and provided reasons why it is worth suggesting.

(3) Please revisit the argued roles of myoglianin in the story, in light of its effects as an inhibitor of JH production, repressing the expression of JHAMT, as has been reliably demonstrated in hemimetabolan species (DOI: 10.1073/pnas.1600612113 and DOI: 10.1096/ fj.201801511R).

Our appreciation to the reviewer. We are more explicit about the relationship between JH and myo.

Minor points(4) Please keep the consistency of the scientific binomial nomenclature for the species mentioned. For example, read "Manduca sexta" (in italics) at the first mention, and then "M. sexta" (in italics) in successive mentions (instead of reading "Manduca" on page 17, and then "Manduca sexta" on page 18, for example). The same for "*Drosophila*" ("*Drosophila melanogaster*" first, and then "*D. melanogaster*"), "Thermobia" ("Thermobia domestica" first, and then "T. domestica"), etc. In the figure legends, I recommend using the complete name: Thermobia domestica, in the main heading.

Where there is no possibility of confusion, we intend to use Thermobia, rather than T. domestica, etc. We think that it is easier for a non-specialist to read and it is commonly done in endocrine papers.

(5) There is no purpose in evolution and biological processes. Thus, I suggest avoiding expressions that have a teleological aftertaste. For example (capitals are mine), on p. 3 "appears to have been extended into postembryonic life where it acts TO antagonize morphogenic and allow the maintenance of a juvenile state".

We have tried to avoid teleological wording.

(6) The title "The embryonic role of juvenile hormone in the firebrat, Thermobia domestica, reveals its function before its involvement in metamorphosis" contains a redundancy ("role" and "function"), and an apparent obviousness ("before its involvement in metamorphosis"). I suggest a more straightforward title. Something like "Juvenile hormone plays developmental functions in the embryo of the firebrat Thermobia domestica, which predate its status quo action in metamorphosis".

As noted above, we are retaining the title since it has already been cited.

(7) Page 2. "The transition from larva to adult then occurred through a transitional stage, the pupa, thereby providing the three-part life history diagnostic of the "complete metamorphosis" exhibited by holometabolous insects (reviews: Jindra, 2019; Truman & Riddiford, 2002, 2019)". I suggest adding the reference ISBN: 9780128130209 9 7 8 - 0 - 1 2 - 8 1 3 0 2 0 - 9, as the most comprehensive and recent review on complete metamorphosis.

Done

(8) Page 3. "These severe developmental effects suggest that the developmental role of JH in insects was initially CONFINED to the embryonic domain" (capitals are mine). This appears contradictory with the observations of Watson, 1967, on the relationships between the apparition of scales and JH, mentioned shortly before by the authors.

This is explained in the Discussion. Although JH can suppress scale appearance in the J4 stage, we have not been able to show that scales appearance is caused by changes in the juvenile JH titer.

(9) Page 4. "we measured JH III levels during Thermobia embryogenesis at daily intervals starting at 5 d AEL". Why not before, like in the case of ecdysteroids? The authors might perhaps argue that the levels of Kr-h1 expression are consistently low from the very beginning, according to Fernandez-Nicolas et al, 2022 (reference cited later in the manuscript).(10) Page 4. "Ecdysteroid titers through embryogenesis and the early juvenile instars were measured using the enzyme immunoassay method (Porcheron et al., 1989) that is optimized for detecting 20-hydroxyecdysone (20E)". The antibody generated by Porcheron (and now sold by Cayman) recognizes ecdysone and 20-hydroxyecdysone alike. But that's not relevant here. I would refer to "ecdysteroids" when mentioning measurements. Also in figure 2B (and "juvenile hormone III" without the formula, in Panel A, for harmonization). And I would not expand on specifications, like those at the beginning of page 5, or towards the end of page

We thank the reviewer for this important correction.

(12) ("the fact that we detected only a slight rise in ecdysteroids at this time (Fig 2B) is likely due to the assay that we used being designed to detect 20E rather than ecdysone").

Omitted.

(11) Page 5. "Low levels of Kr-h1 transcripts were present at 12 hr after egg deposition, but then were not detected until about 6 d AEL when JH-III first appeared". There is a very precise Kr-h1 pattern in Fernandez-Nicolas et al. 2023 (reference mentioned later in the manuscript).(12) Page 5. "notably myoglianin (myo), have become prominent as agents that promote the competence and execution of metamorphosis in holometabolous and hemimetabolous insects (He et al., 2020; Awasaki et al., 2011)". See my note 3 above.

The myoglianin issue has been revised.

(13) Page 5. "a drug that suppresses JH production". Rather, "a drug that destroys the JH producing tissues". Why the way, do the authors know when the CA are formed in T. domestica embryo development?

We prefer to keep our original wording. There have been some cases in which precocene has blocked JH production but did not kill the CA cells. We do not have observations that show that 7EP kills the CA cells in Thermobia embryos.

(14) Page 5. "subsequent treatment with a JHm". I would say here that the JHm is pyriproxyfen, not on page 6 or page 7. Thus, to be consistent, after the first mention of "pyriproxyfen (JHm)" on page 5, I'd consistently use the abbreviation "JHm".(15) Page 9. "Limb loss in such embryos was often STOCHASTIC, i.e., in a given embryo some limbs were completely lost while others were maintained in a reduced state" (capitals are mine). The meaning of "stochastic" is random, involving a random variable; it is a concept usually associated to probability theory and related fields. I suggest using the less specialized word "variable", since to ascertain that the values are really stochastic would require specific mathematical approaches.

We are still using stochastic because the loss is random.

(16) Page 10. "(9E). Indeed, the JH treatment redirects the molt to be more like that to the J2 stage, rather than to the E2 ( = J1) stage". Probably too assertive given the evidence available (see my points 1 and 2 above).

We do not see a problem with our conclusion. In response to the JHm treatment, the embryo produced a smooth, rather than a “pebbly” cuticle, failed to make the J1-specific egg tooth, and attempted to make cuticular lenses (a J2 feature). This ability of premature JH exposure to cause embryos to “skip” a stage is also seen in locusts (Truman & Riddiford, 1999) and crickets (Erezyilmaz et al., 2004). The JHm treatment resulted in the production of smooth cuticle, lack of a hatching tooth, and an attempt to make cuticular lenses.

(17) Page 11. "early JHM treatment", read "early JHm treatment".

Corrected

(18) Page 11. "likely. A target of JH, and likely Kr-h1, in Thermobia is myoglianin...". Please see my notes 1, 2, and especially 3, above.

This has been revised

(19) Page 13. "the locust, Locusta americana (Aboulafia-Baginshy et al.,1984)". Please read "the locust, Locusta migratoria (Aboulafia-Baginshy et al.,1984)".

Corrected

(20) Page 13 "Acheta domesticus" three times. The correct name now is "Acheta domestica", after harmonizing the declension of the specific name with the generic one. See additionally my note 4 above.

Acheta domesticus has been used in hundreds (thousands?) of papers since it was originally named by Linnaeus. We will continue to use it.

(21) Page 15, "(also called the vermiform larva (Bernays, 1971)) redirects embryonic development to form an embryo with proportions, cuticular pigmentation, cuticular sculpturing and bristles characteristic of a nymph, while pronymph modifications, such as the cuticular surface sculpturing (Bernays, 1971)". The reference "Bernays, 1971" is indeed "Bergot et al., 1971".

There was a mistake in the references. The Bernays reference was omitted from the revised Discussion

(22) Page 16. "Since JH also induces Kr-h1 in embryos of many insects, including Thermobia". I'm not sure that this has been studied in many insects. In any case, any reference would be useful.(23) Page 17. "Tribolium casteneum". Please read "Tribolium castaneum".

Changed

(24) Page 17. "...results in a permanent larva that continues to molt well after it has surpassed its critical weight (He et al., 2019)". The paper of He et al., 2019 is preceded by two key papers that previously demonstrate (and in hemimetabolan insects) that myoglianin is a determining factor in the preparation for metamorphosis: (DOI: 10.1073/pnas.1600612113 and DOI: 10.1096/ fj.201801511R). See my note 3 above.

Corrected in revision

(25) Page 18. "These persisting embryonic primordia join the wing primordia in delaying their morphogenesis into postembryonic life". This reader does not understand this sentence.

Made clearer in the revision.

(26) Page 18. "is first possible in the commercial silkworm (Daimon et al., 2015)". Please mention the scientific Latin name of the species, Bombyx mori.(27) Page 19. "The functioning of farnesol derivatives in growth versus differentiation control extends deep into the eukaryotes.../... this capacity was eventually exploited by the insects to provide the hormonal system that regulates their metamorphosis". This information appears quite out of place.

We have retained this point.

(28) Page 21. Heading "Hormones". I suggest using the heading "Bioactive compounds", as neither pyriproxyfen nor 7-ethoxyprecocene are hormones.

Done

(29) Page 29, legend of figure 1. "Photomicrographs" is somewhat redundant. The technical word is "micrographs". "Thermobia domestica" appears in the explanation of panel C, but this is not necessary, as the name appears in the main heading of the legend.

Done

(30) Page 30, legend of figure 2. Panel B, see my comment 10 above. Why embryonic age is expressed in % embryo development in panel C (and in days in panels A and B)?

All have been converted to days AEL

(31) Page 35, legend of figure 5. "Photomicrograph" see my note 28 above.

Done

(32) Page 40, figure 10. In panel A, the indication of the properties of JH is misleading. The arrow going to promoting differentiation and maturation is OK, but the repression sign that indicates suppression of morphogenetic growth and cell determination seems to suggest that JH has retroactive effects. In panel B, I suggest to label "Flies" instead of "Higher Diptera", which is an old-fashioned term. In any case, see my general comments 1 and 2, above, about speculation.

Figure has been completely revised

(33) Figure 11. See my general comments 1 and 2, above, about speculation.

Figure has been revised

**Reviewer #3 (Public Review):**
Summary:In this manuscript, the authors use inhibitors and mimetics of juvenile hormone (JH) to demonstrate that JH has a key role in late embryonic development in Thermobia, specifically in gut and eye development but also resorption of the extraembryonic fluid and hatching. They then exogenously apply JH early in development (when it is not normally present) to examine the biological effects of JH at these stages. This causes a plethora of defects including developmental arrest, deposition of chitin, limb development, and enhanced muscle differentiation. The authors interpret these early effects on development as JH being important for the shift from morphogenetic growth to differentiation - a role that they speculate may have facilitated the evolution of metamorphosis (hemi- and holo-metaboly). This paper will be of interest to insect evo-devo researchers, particularly those with interests in the evolution of metamorphosis.Strengths:The experiments are generally conducted very well with appropriate controls and the authors have included a very detailed analysis of the phenotypes.The manuscript significantly advances our understanding of Thermobia development and the role of JH in Thermobia development.The authors interpret this data to present some hypotheses regarding the role of JH in the evolution of metamorphosis, some aspects of which can be addressed by future studies.Weaknesses:The results are based on using inhibitors and mimetics of JH and there was no attempt to discern immediate effects of JH from downstream effects. The authors show, for instance, that the transcription of myoglianin is responsive to JH levels, it would have been interesting to see if any of the phenotypic effects are due to myoglianin upregulation/suppression (using RNAi for example). These kinds of experiments will be necessary to fully work out if and how the JH regulatory network has been co-opted into metamorphosis.

We agree completely and should be a feature of future work.

The results generally support the authors' conclusions. However, the discussion contains a lot of speculation and some far-reaching conclusions are made about the role of JH and how it became co-opted into controlling metamorphosis. There are some interesting hypotheses presented and the author's speculations are consistent with the data presented. However, it is difficult to make evolutionary inferences from a single data point as although Thermobia is a basally branching insect, the lineage giving rise to Thermobia diverged from the lineages giving rise to the holo- and hemimetabolous insects approx.. 400 mya and it is possible that the effects of JH seen in Thermobia reflect lineage-specific effects rather than the 'ancestral state'. The authors ignore the possibility that there has been substantial rewiring of the networks that are JH responsive across these 400 my. I would encourage the authors to temper some of the discussion of these hypotheses and include some of the limitations of their inferences regarding the role of JH in the evolution of metamorphosis in their discussion.

We have tried to be less all-encompassing in the Discussion. The strongest comparisons can be made between ametabolous and hemimetabolous insects and we have focused most of the Discussion on the role of JH in that transition. We still include some discussion of holometabolous insects because the ancestral embryonic functions of JH may be somehow related to the unusual reappearance of JH in the prepupal period. We have reduced this discussion to only a few sentences.

**Reviewer #3 (Recommendations For The Authors):**
(1) The overall manuscript is very long (especially the discussion), and the main messages of the manuscript get lost in some of the details. I would suggest that the authors move some of the results to the supplementary material (e.g. it might be possible to put a lot of the detail of Thermobia embryogenesis into the supplementary text if the authors feel it is appropriate). The discussion contains a lot of speculation and I suggest the authors make this more concise. One example: At the moment there is a large section on the modification in JH function during the evolution of holo and hemi-metabolous life history strategies. There are some interesting ideas in this section and the authors do a good job of integrating their findings with the literature - but I would encourage the authors to limit the bulk of their discussion to the specific things that their results demonstrate. E.g. The first half of p17 contains too much detail, and the focus should be on the relationship with Thermobia (as at the bottom of p17).

Section has been revised and is more focused

(2) I would also suggest a thorough proofread of the manuscript, I have highlighted some of the errors/points of confusion that I found in the list below - but this list is unlikely to be exhaustive.We appreciate catching the errors. Hopefully the final version is better proofed.(3) It might be me, but I found the wording in the second half of the abstract a bit confusing. Particularly the statement about the redeployment of morphogen systems - could this be stated more clearly?

Abstract has been revised.

(4) Introductiona. "powered flight" rather than 'power flight'

Done

b. 'brought about a hemimetabolous lifecycle' implies causality which hasn't been shown and directionality to evolution - suggest 'facilitated the evolution of a hemi...". Similar comment for 'subsequent step to complete metamorphosis'.c. Bottom of p2 - unclear whether you are referring to hemi- holo- or bothd. Suggest removing sentence beginning "besides its effects..." as the relevance of the role of JH in caste isn't clear.

Kept sentence but removed initial clause

e. State that Thermoia is a Zygentoma.

Done

f. Throughout - full species names on first usage only, T. domestica on subsequent usages.

We will continue to use genus names for the reason given above.

Gene names e.g. kr-h1 in italics.

g. 'antagonise morphogens"? rather than 'antagonise morphoentic'.

Done

(5) Resultsa. Unclear why drawings are provided rather than embryonic images in Fig. 1A

We think that the points can be made better with diagrams.

b. Top of p4, is 'slot' the correct word?

Corrected

c. Unclear why the measurements of JHIII weren't measured before 5 days AEL, especially given that many of the manipulative experiments are at earlier time points than this. I appreciate that, based on kr-h1, levels that JHIII is also likely to be low.d. Reference for the late embryonic peak of 20E being responsible for the J2 cuticle?

Clarified that this is an assumption

e. Clarify "some endocrine related transcripts" why were these ones in particular picked? Kr-h1 is a good transcriptional proxy for JH and Met is the JH-receptor, why myoglianin and not some of the other transcriptional proxies of neuroendocrine signalling?

Hopefully, the choice is clearer.

f. Fig 2C rather than % embryo development for the gene expression data please represent this in days (to be consistent with your other figures).

It is now consistent with other parts of figure.

g. In Fig. 3 the authors do t-tests, because there are three groups there needs to be some correction for multiple testing (e.g. Bonferroni) can the authors add this to the relevant methods section?

We think that pair-wise comparisons are appropriate.

h. Fig. 3 legend: you note that you treat stage 2 juveniles with 7EP - I couldn't tell what AEL this corresponded to.

This is after hatching so AEL does not apply.

i. Top of p7 'deformities' rather than 'derangements'?

Done

j. Regarding the dosage effects of embryonic abnormalities - it would be good to include these in the supp material, as it convinces the reader that the effects you have seen aren't just due to toxicity.

It is not clear what the objection is.

k. Bottom of p7 'problematic' not 'problematical'

Done

l. P8 Why are the clusters of Its important? - provide a bit more interpretation for the reader here.

This is clear in the revised version.

m. P9 Why is the modulation of transcription of kr-h1, met, and myo important in this context

Explained

n. P9 'fig. 7F'? there is no Fig. 5F

Thanks for catching the typo.

o. Fig. 7B add to the legend which treatment the dark and light points correspond to.

We think it is obvious from the labeling on Fig 7B.

(6) Discussion:a. What do we know about how terminal differentiation is controlled in non-insect arthropods? Most of the discussion is focused on insects (which makes sense as JH is an insect-specific molecule), but if the authors are arguing the ancestral role of JH it would be useful to know how their findings relate to non-insect arthropods.

We have not been able to find any information about systemic signals being involved in non-insect arthropods.

b. There is no Fig. 5E (are they referring to 7E?)

Yes, it should have been Fig. 7E.

c. Is myoglianin a direct target of JH in other species?

Other reports are in postembryonic stages and show that myoglianin suppresses JH production. Our paper is the first examination in embryos and we find that the opposite is true – i.e., that JH treatment suppresses myoglianin production. We suspect that these two signaling systems are mutually inhibitory. It would be interesting to see whether treatment of a post-critical weight larva with JH (which would induce a supernumerary larval molt) would also suppress myoglianin production (as we see in Thermobia embryos).

d. P12 What is the evidence that JH interacts with the first 20E peak to alter the embryonic cuticle?

We are not sure what the issue is. The experimental fact is that treatment with JH before the E1 ecdysteroid peak causes the production of an altered E1 cuticle. We are faced with the question of why is this molt sensitive to JH when the latter will not appear until 3 or 4 days later? A possible answer is that the ecdysone response pathway has a component that has inherent JH sensitivity. The mosquito data suggest that Taiman provides another link between JH and ecdysone action

e. Top of p13 - this paragraph can be cut down substantially. Although this is evidence that JH can alter ecdysteriods - it is in a species that is 400 my derived from the target species. Is it likely to be the exact same mechanism? I would encourage the authors to distil and retain the most important points.

This paragraph has been shortened and focused.

f. Bottom of p13 - what does this study add to this knowledge?

The response of Thermobia embryos to JH treatment is qualitatively the same as seen in other short germband embryos. This similarity supports the assumption that the same responses would have been seen in their last common ancestor.

g. P19 the last paragraph in the conclusions is really peripherally relevant to the paper and is a bit of a stretch, I would encourage the authors to leave this section out.

We agree that it is a stretch. JH and its precursor MF are the only sesquiterpene hormones. How did they come about to acquire this function? We think it is worth pointing out the farnesol metabolites have been associated with promoting differentiation in various eukaryotes. An ancient feature of these molecules in promoting (maintaining?) differentiation may have been exploited by the insects to develop a unique class of hormones. It is worth putting the idea out to be considered.

h. P19 "conclusions" rather than 'concluding speculations'.

Changed as suggested.

Methods:It is standard practice to include at least two genes as reference genes for RT-qPCR analysis (https://doi.org/10.1186/gb-2002-3-7-research0034, https://doi.org/10.1373/clinchem.2008.112797) If there are large-scale differences in the tissues being compared (e.g. as there are here during development) then more than two reference genes may be required and a reference gene study (such as https://doi.org/10.3390%2Fgenes12010021) is appropriate. Have the authors confirmed that rp49 is stably expressed during the stages of Thermobia development that they assay here?

We have explained our choice in the Methods.